



# Dimensions of Marine Phytoplankton Diversity

Stephanie Dutkiewicz[1,2], Pedro Cermeno[3], Oliver Jahn[1], Michael J. Follows[1], Anna E. Hickman[4], Darcy A.A. Taniguchi[5], Ben A. Ward[4]

1. Department of Earth, Atmospheric and Planetary Sciences, Massachusetts Institute of Technology, Cambridge, MA, 02139, USA
2. Center for Climate Change Science, Massachusetts Institute of Technology, Cambridge, MA, 02139, USA
3. Institut de Ciencies del Mar, CSIC, 08003 Barcelona, Spain
4. Ocean and Earth Sciences, University of Southampton, National Oceanography Centre Southampton, Southampton, SO14 3ZH, United Kingdom.
5. Department of Biological Sciences, California State University San Marcos, San Marcos, CA, 92096 USA

*Correspondence to*: Stephanie Dutkiewicz (stephd@mit.edu)

**Abstract.** Biodiversity of phytoplankton is important for ecosystem stability and marine biogeochemistry. However, the large scale patterns of diversity are not well understood, and are often poorly characterized in terms of statistical relationships with environmental factors (e.g. latitude, temperature, productivity). Here we use ecological theory and a global trait-based ecosystem model to provide mechanistic understanding of patterns of phytoplankton diversity. Our study suggests that phytoplankton diversity across three dimensions of trait space (size, biogeochemical function, and thermal tolerance) is controlled by a disparate combinations of drivers: the supply rate of the limiting resource, the imbalance in different resource supplies relative to competing phytoplanktons' demands, size-selective grazing, and transport by the moving ocean. Using sensitivity studies we show that each dimension of diversity is controlled by different drivers. Models including only one (or two) of the trait dimensions will have different patterns of diversity than one which incorporates another trait dimension. We use the results of our theory/model exploration to infer the controls on the diversity patterns derived from field observations in meridional transects of the Atlantic and to explain why different taxa and size classes have differing patterns. These results suggest that it is unlikely that any single or even combination of environmental variables will be able to explain patterns of diversity.

## 1 Introduction

Phytoplankton are an extremely diverse set of microorganisms spanning more than 7 orders of magnitude in cell volume (Beardall et al., 2008) and an enormous range of cell morphologies, biochemical functions, elemental requirements and trophic strategies. This range of traits play a key role in regulating the biogeochemistry of the ocean (e.g. Cermeno et al., 2008; Fuhrman 2009) including the export of organic matter to the deep ocean (Guidi et al, 2009), which is critical in oceanic carbon sequestration and contributes to modulation of atmospheric $CO_2$ levels and climate. Biodiversity is also important for





the stability of the ecosystem structure and function (e.g. McCann 2000; Ptacnik et al 2008), though the exact nature of this relationship is still debated. Studies suggest that diversity loss appears to coincide with a reduction in primary production rates and nutrient utilization efficiency (Cardinale et al., 2011; Reich et al., 2012), thereby altering the functioning of ecosystems

and the services they provide. It is clear diversity is important, but what controls diversity still remains an elusive problem.

Numerous studies have attempted to understand or predict observed patterns of biodiversity of marine phytoplankton by correlating with environmental factors (see e.g. Hillebrand and Azovsky, 2001; Hillebrand, 2004; Irigoien et al. 2004; Smith et al, 2007; Rodriguez-Ramos et al 2015; Powell et al, 2017; Righetti et al. 2019). The metabolic theory of ecology posits that temperature could control the probability of mutation and speciation leading to more diversity at higher temperatures (see e.g.

Allen et al 2007). A recent study, on the other hand, suggests a unimodal statistical relationship between diversity and temperature (Righetti et al., 2019). Studies have also proposed a latitudinal dependence of diversity (e.g. Chust et al 2012), though the shape of that dependence is unclear. Chaudary et al 2016 for instance suggests a bimodal distribution, and a study of Cenzioc fossil records suggest that diversity of diatoms may actually have increased towards the poles (Powell and Glazier, 2017). However, Rodriguez-Ramos et al (2015), found little evidence of a relationship between nano- and micro-phytoplankton

diversity and either temperature or latitude after enforcing consistency of data sets. Additionally, there is evidence suggest that increased dispersal (up to a point) could increase diversity (Matthiessen and Hillebrand, 2006) and diversity was related to meso-scale features in a study in the North Atlantic (Mousing et al, 2016).

There has been a debate as to how productivity links to diversity (see e.g. review by Smith, 2007). Again, by standardizing data sets to correct for differences in sampling efforts only weak (or no) correlation between phytoplankton diversity and

productivity emerges from global data sets (Cermeno et al 2013; Rodriguez-Ramos et al., 2015) suggesting that previously reported connections might be skewed by sampling biases (Cermeno et al, 2013). A recent study of genomic data for the full planktonic community also showed little variance could be explained by environmental factors, including nutrient concentrations (Lima-Mendez et al., 2015). However biotic interactions had much better predicative power. The importance of top down control has been suggested by the experiments of Worm et al (2002). Multiple factors appear to be likely important,

but correlations with multiple co-occurring environmental factors do not satisfactorily explain diversity patterns (e.g. Rodriguez-Ramos et al 2015; Lima-Mendez et al 2015). There remains no holistic understanding of phytoplankton diversity and its drivers.

Recent theoretical work (e.g. Vallina et al 2014b; 2017; Treseleer et al 2015) suggest that breaking diversity down into traits can be useful. Vallina et al (2017) also suggested that a variety of traits respond differently to environmental factors. The

importance of multiple phytoplankton traits in setting community structure has previously been expounded (e.g. Litchman et al 2010, Acevedo-Trejos et al., 2015). Theory and models have considered several different phytoplankton traits and environmental drivers to explain diversity. In one study, different temperature dependencies and nutrient affinity trade-offs allowed phytoplankton to have similar lowest subsistence nutrient requirements (as described in Tilman, 1977; 1982) that allowed sustained co-existence (Barton et al 2010). Other studies explored the importance of top-down control (Prowe et al,

2012; Vallina et al., 2014a, Ward et al 2014). A positive relationship between diversity and productivity was found when a





model captured only different size classes, but no temperature differences (Ward et al., 2012; 2014). A series of studies also showed the importance of dispersal for diversity (Levy et al 2014), that mesoscale features enhanced diversity (Levy et al. 2014; 2015; Clayton et al 2013), and that hot spots of diversity occurred in regions of high mixing (Clayton et al 2013).

In this study we seek to disentangle the multiple, sometimes conflicting, results from models and observational studies, and seek to explain at least some of the controls on diversity. We employ ecological theories and a trait-based global model. We use observed patterns of diversity along meridional transects of the Atlantic as motivation, and as illustration of the utility of this study. By using model and theory, we explore the mechanistic drivers of the modelled diversity. By using a model, we can conduct sensitivity experiments to test the intuition that theoretical framework provides. However, on a cautionary side, this study tells us about the diversity in the model world. Though our model is complex, it still missed many of the traits of the real

ocean microbial communities.

This study synthesises much of the understanding that we have gained through previous models and theoretical studies (e.g. Dutkiewicz et al, 2009,2012, 2014; Ward et al 2013; 2014; Levy et al 2014). What is unique here is bringing these all together, addressing disparities in previous work and providing insight into the multiple interacting mechanisms that drive diversity. We find that this can only be done by acknowledging that diversity along different axes of traits (e.g. size, biogeochemical function,

thermal norms) each have their own set of drivers. And this is turn suggests that no single or combined set of environmental variables will be able to explain patterns of diversity in the real ocean.

## 2. Methods

### 2.1 Atlantic Meridional Transect (AMT) Observations

As an illustrative example from field observations, we used data of species composition, abundance and cell size in the range of nano- and micro-phytoplankton from samples collected in marine pelagic ecosystems. Results from the coccolithophore and diatom diversity from this dataset have previously been shown in Cermeno et al (2008). The data come from transects sampled during September to October 1995 (AMT-1), April to May 1996 (AMT-2), September to October 1996 (AMT-3), and April-May 1997 (AMT4). The courses of these cruises crossed the same regions of the Atlantic Ocean by a similar route. At each

station, 2 replicate seawater samples were preserved, one with 1 % buffered formalin (to preserve calcite structures) and the other with 1 % final concentration Lugol's iodine solution. After sedimentation of a sub-sample for 24 h (Utermöhl's technique), cells were measured and counted with an inverted microscope at x187, x375 and x750 magnifications to cover the full ensemble of nano- and micro-phytoplankton, and identified to the lowest possible taxonomic level (usually species level). The volume of water samples used for sedimentation varied between 50 and 256 ml, according to the overall abundance of

phytoplankton as shown by the fluorometer. At least 100 cells of each of the more abundant species were enumerated. Cell volume was calculated by assigning different geometric shapes that were most similar to the real shape of each phytoplankton species. A mean cell volume was assigned for each phytoplankton species. Cells were separated into diatoms, coccolithophore



and dinoflagellate groups. Here these data are used to determine total richness (number of co-existing species) of all the nano and micro-eukaryotes (Fig 1a), but also species richness within diatom, dinoflagellate and coccolithophore groups (Fig 1b), as

well as number of species in three size groups (2-10 µm, 10-20 µm, >20 µm, Fig 1c).

## 2.2. Numerical Model

The model follows from Dutkiewicz et al (2015a) in terms of biogeochemistry, plankton interactions, and transmission of light as described by the tables and equations of that paper. However, the types of phytoplankton and zooplankton differ in that they

include greater diversity. Here we briefly provide an overview of the model, and some more detailed descriptions of the more complex ecosystem. More details and table of pertinent parmaters can be found in the Supplemental material and the full set of equations and parameters can be found in Dutkiewicz et al (2015a).

The biogeochemical/ecosystem model resolves the cycling of carbon, phosphorus, nitrogen silica, iron, and oxygen through inorganic, living, dissolved and particulate organic phases. The biogeochemical and biological tracers are transported and

mixed by the MIT general circulation model (MITgcm, Marshall *et al.*, 1997) constrained to be consistent with altimetric and hydrographic observations (the ECCO-GODAE state estimates, Wunsch and Heimbach, 2007). This three-dimensional configuration has coarse resolution (1°×1° horizontally) and 23 levels ranging from 10m in the surface to 500m at depth.

We use a complex marine ecosystem that incorporates many phytoplankton types that can be described in 3 "dimensions" of trait space (schematically shown in Fig 2): size, biogeochemical function, and temperature tolerance. Within the "size"

dimension we include 16 size classes spaced uniformly in log space from 0.6 µm to 228 µm equivalent spherical diameter (ESD). Within the "biogeochemical function" dimension we resolve diatoms (that utilize silicic acid), coccolithophores (that calcify), mixotrophs (that photosynthesize and graze on other plankton), nitrogen fixing cyanobacteria (diazotrophs), and pico-phytoplankton. We resolve 4 size classes of pico-phytoplankton (from 0.6 to 2 µm ESD), 5 size classes of coccolithophores and diazotrophs (from 3 to 15 µm ESD), 11 size classes of diatoms (3 to 155 µm ESD), and 10 mixotrophic dinoflagellates

(from 7 to 228 µm ESD). Additionally, we resolve a "temperature norm" trait axis, where phytoplankton growth rates are defined over a specific range of temperatures (Fig 3) by an empirically motivated function (e.g. Thomas et al, 2012, Boyd et al, 2013). We include 10 different norms. Thus for any size class within a functional group there are 10 different unique phytoplankton types (as demonstrated schematically in Fig 2) with different range of temperatures over which the cells will grow. Warmer adapted types are assumed to grow faster as suggested empirically (Eppley, 1972, Bissenger et al, 2008) and

from enzymatic kinetics (Kooijman, 2000). In total we resolve 350 phytoplankton "types" within 16 size classes, 5 biogeochemical functional groups, and 10 temperature norms.

Phytoplankton parameters influencing maximum growth rate, nutrient affinity, grazing, and sinking are parameterized as a power function of cell volume: $aV^b$ (following Ward et al., 2012; see Supplemental text S1.2 and Table S1). Thus many size classes can be described by just two coefficients (a,b) per parameter. Maximum growth rate is parameterized (i.e. the a,b in

the above equation) as distinct between functional groups (as suggested by observations in Fig 4a, see also Buitenhuis et al



2008; Sommer et al 2017). The smallest diatoms (3um) have the highest maximum growth rates. Plankton smaller than 3um have an increase of growth rate with size, and those larger than 3um have a decrease of growth rate with size. This unimodal distribution has been observed (e.g. Raven 1994; Bec et al 2008; Finkel et al 2010; Maranon et al 2013; Sommer et al 2017) and explained as a tradeoff between replenishing cell quotas versus synthesizing new biomass (Verdy et al., 2009; Ward et al

2017). There are also specific differences between functional groups in cell elemental stoichiometry, and palatability to grazers (diatoms and coccolithophores, with their hard surface covering deter grazers, see e.g. Monteiro et al., 2016, Pančić et al., 2019). The smallest phytoplankton have the highest affinity for nutrients (Edwards et al., 2012) as a result of the lowest surface to volume ratio in larger cells (Kiorboe 1993, Raven, 1994).

The model includes spectral irradiances, and each functional group has different spectra for absorption (as a result of group

specific accessory pigments) and scattering of light. The absorption spectra are flatter with larger sizes following Finkel et al (2000) to account for self-shading, and scattering spectra are also influenced by size following Stramski et al (2001) (see Supplemental text S1.3, Supplemental Fig S1). The simulation uses Monod kinetics, and C:N:P:Fe stoichiometry are constant over time (though differ between phytoplankton groups). However, Chl-a for each phytoplankton types varies in time and space depending on light, nutrient and temperature conditions following Geider et al (1998). Following empirical evidence,

mixotrophic dinoflagellates are assumed to have lower maximum photosynthetic growth rates than other phytoplankton of the same size (Tang, 1995; Fig 4a) and lower maximum grazing rate heterotrophic dinoflagelletes of the same size (Jeong et al., 2010, Supplemental Fig S2).

We resolve 16 size classes of zooplankton (from ESD 6.6um to 2425um) that graze on plankton (phyto- or zoo-) 5 to 20 times smaller than themselves, but preferentially 10 times smaller (Hansen et al., 1997; Kiorboe et al 2008, Schartau et al 2010).

Maximum grazing rate is a function of size (following Hanson et al. 1997), though the four smallest grazers are assumed to have the same maximum grazing rates (Supplemental Fig S2). Here the smallest grazers do not have a clear difference in grazing related to size (following the data compilation of Taniguchi et al, 2014). We use a Holling III grazing function (Holling, 1959). Sensitivity studies with a Holling II parameterization show that the results here are not sensitive to this choice.

We perform a "default" simulation (EXP-0) for 10 years. The ecosystem quickly (within 2 years) reaches a quasi-steady state.

Here we show results from the 5[th] year of the simulation, but note that the patterns of biogeochemical and ecologically relevant output, and diversity are not significantly different if we instead used the 10[th] year. We also conducted a series of sensitivity experiments, where we alter either physical or ecosystem assumptions to provide evidence for the controls of diversity (Table 1).

We will primarily discuss diversity in term of "richness" defined here as the number phytoplankton types that co-exist at any

location above a threshold. We, in particular, look at the annual mean of the instantaneous surface richness (though see Supplemental for examples with depth). Technically we use a threshold value ($10^{-5}$ mmolC/m$^3$) to determine if a type is in existence at any spot. This value would convert to about 10 *Prochlorococcus* cells/ml (typical oligotrophic waters are above $10^3$ cells/ml), or only a tiny fraction ($10^{-4}$) of a larger diatom cell/ml. Lower than this value is assumed numerical noise. The value of richness can be altered depending on the threshold chosen, but the patterns and results discussed below remain robust.





## 3. Results

### 3.1. Diversity Observations along the AMT.

The four Atlantic Meridional Transect (AMT) cruises provide a large-scale consistent dataset of phytoplankton diversity including microscopic counts of diatoms, coccolithophores and dinoflagellates. Such microscopic measurements depict species richness patterns of abundant taxa, but miss much of the rare biosphere. This dataset shows distinct large scale patterns (Fig 1a), with high richness (as determined by number of co-existing species, see methods) on the northern edge of the Southern Ocean, in the Canary upwelling, low richness in the subtropical gyres, and slightly elevated richness in the equatorial region. However, the patterns of richness are very different if we look only within a single functional group (e.g. diatoms, Fig 1b) or within a specific size class (Fig 1c). Diatoms exhibit higher diversity in the Southern Ocean than the other functional groups, while the diversity of coccolithophores and dinoflagellates is much more uniform across the transects. Among size classes, the smallest size category (2-10um) has the highest diversity, while there is lower and more regionally varying diversity in the larger size categories, with some regions having none of the largest size class (>20um). This suggests that the controlling mechanism(s) on, for instance, diatom diversity is different to those controlling coccolithophore diversity, which also differs to what determines the diversity within different size classes. Indeed, modelling and theoretical work (e.g. Vallina et al 2014b; 2017; Terseleer et al 2014) have suggested that breaking diversity down into traits can be insightful. Thus, a starting point of our study is to separate out different dimensions of diversity.

### 3.2. Numerical Model

Model development was guided by evaluating against a range of in situ and satellite-derived observations as in Dutkiewicz et al (2015a). The model captures the patterns of low and high Chl-a seen in the satellite estimate (Supplemental Fig S3), though underestimates the Chl-a in the subtropical gyres. The coarse resolution of the model does not capture important physical processes near coastlines, and lack of sedimentary and terrestrial supplies of nutrients and organic matter lead to Chl-a being too low in these regions. The underestimation of Chl-a in the gyres is also seen when comparing the model to the observations of surface Chl-a along the Atlantic meridional transects (AMT, Fig 5). The model does capture the drawdown of nutrients in the gyres and the large increase of nutrient concentrations in the Southern Ocean. However, the model over-estimates the amount of silicic acid in this ocean, possibly a reflection of Si:C of the model diatoms being too low in the region. We also compare the model biomass of diatoms, coccolithophores and dinoflagellates along the AMT. Though note that the conversion from cell counts to biomass in the observations (see Section 2.1) has a significant uncertainty. The model captures the much lower biomass of diatoms in the subtropical gyres than the other two functional groups, and higher in the Southern Ocean. Coccolithophore biomass is too low in the Southern Ocean in the model, likely due to the smallest diatom being parameterized as too competitively advantaged.





The model individual types have plausible ranges (4 representative species shown in Supplemental Fig 4) given distributions determined from thermal niches (e.g. Thomas et al., 2012) and statistical techniques from the sparse observations (e.g. Barton et al. 2016). The model captures biomass in almost all size classes (Fig 6, Supplemental Figs5a), though the largest size classes are likely underestimated. Traits not included in the model (e.g. buoyancy regulation, chain formation, symbiosis) are possibly

more important for maintaining these large size classes. The model has biomass in all temperature norms (Fig 6, Supplemental Fig 5b), though with lower biomass in the coldest and warmest adapted suggesting the model parameterization covers an adequate range of norms. However, there are some interesting eliminations (which match observations) such as coldest adapted smallest pico-phytoplankton and diazotrophs, and large warmest-adapted diatoms. The phytoplankton are complemented by a range of size classes of zooplankton (Supplemental Fig S6).

We evaluate the model's ability to capture the size distribution of phytoplankton as derived from satellite products (Fig 7a). Here we capture the ubiquitous pico-phytoplankton and the limitation of the larger size classes to the more productive regions. The model pico-phytoplankton size class Chl-a is potentially slightly too low and the nano size class too high. Though we note that if we set the pico/nano break at the model 5[th] size class (just under 3um) instead at the 4[th] (2um) size class, the relative values are much more in line with the satellite product. We suggest that the satellite product division might not be that exact.

The micro-size class matches in location to the satellite product but is slightly too low as discussed above.

We also compare the model functional group distribution to the latest compilation of observations (Fig 7b, MAREDAT, Buitenhuis et al 2013, and references therein). Though the observations are sparse, we do capture the ubiquitous nature of the pico-phytoplankton, the limited domain of the diazotrophs (including observed lack of diaztrophs in the South Pacific gyre), the pattern of enhance diatom biomass in high latitude, and low in subtropical gyres. We over-estimate the coccolithophore

biomass relative to MAREDAT in many regions, but note that the conversion from cells to biomass in that compilation was estimated to have uncertainties as much as several 100% (O'Brien et al., 2013). The MAREDAT compilation did not include a category for dinoflagellates.

In this manuscript we mostly consider richness, the number of co-existing types (see Section 2.2), as a metric of diversity. This is because the ecological theories we use explain co-existence, rather than evenness. However, we do discuss Shannon Index

(another commonly used metric of diversity) later in the text. We will refer to "total" richness, i.e. the number of co-existing phytoplankton types, out of the 350 initialized in the model, at any location (Fig 8a). Here we specifically look at the annual mean richness in the surface layer which is a good indicator of the diversity within the mixed layer (Supplemental Fig S7). We find lowest richness in the subtropical gyres and highest associated with the western boundary currents.

The model is designed to allow for richness within specific functional groups and size classes. A unique feature about this

study is a comparison to the richness found in the AMT data (Fig 1). The model captures the low and high patterns of the AMT observations, though underestimates the diversity in the subtropical gyres. In these regions it is likely that traits axes (e.g. symbiosis, colony formation etc) not captured in the model provide additional means for phytoplankton to co-exist. Excitingly the model also captures the differences in the diversity within functional groups and in size classes. Diatoms have much larger diversity in the Southern Ocean than the other functional groups, while coccolithophores and mixotrophic dinoflagellates



diversity is much more uniform across the transect. The model captures the much higher diversity within the smallest size category (2-10um) and the lower and much more regionally varying diversity in the larger size category, including the lack of diversity in the largest size class (>20um) in the subtropical gyres.

It is instructive to also consider richness along each of the dimensions of trait space. The number of size classes (irrespective

of functional group or thermal norm) that co-exist in any location will be referred to as size class diversity (Fig 8b). We find that in high latitudes and along the equator, many size classes are present, while in the subtropical gyres only few, small-sized classes survive (Fig 7a, Supplemental Fig S5a).  We find that there are different patterns of richness when looking along the two other axes of traits (Fig 8c,d; Supplemental Fig S5b,c). Richness of biogeochemical functional groups is highest in the mid-latitudes, strongly linked to the distributions of diazotrophs (Fig 7b, Supplemental Fig S5b). On the other hand, the

diversity within temperature norms is maximum in the western boundary currents, in particular the Gulf Stream and Kuroshio, and high in coastal upwelling regions (e.g. off Peru and Canary) and along the northern boundary of the Southern Ocean.
The total richness is a complex integral function (i.e. multiplicative) of the three different trait dimensions. We find that some trait dimensions are more (or less) important in different regions.  For instance, thermal norm richness leads to the total richness hotspots (Fig 8a) in the western boundary currents and coastal upwelling regions. While reduction in functional groups and

thermal norms counteract the increase in size classes in the Southern Ocean, all three dimensions together lead to the lowest total richness captured in the middle of the subtropical gyres.

## 4. Understanding the Dimensions of Diversity: Model and Theoretical Framework

None of these three dimensions can, in isolation, explain the controls on the total richness. Nor can we a priori understand the total richness. By using ecological theories and a series of sensitivity experiment (Table 1), we can understand the mechanisms

setting the different dimensions of diversity individually. Here we step through each of the dimensions.
The theoretical frameworks are presented in the Appendix and are informed from the seminal work of Tilman (1977, 1982) and Armstrong (1994).  Resource competition theory (Tilman 1977, 1982) has been extensively used in theoretical and experimental studies (e.g. Sommer 1986; Grover, 1991a, 1991b; Huisman et al, 1994; Schade et al 2005; Miller et al., 2005; Wilson et al., 2007; Agawin et al 2007; Snow et al 2015) as well as linking to numerical models (Dutkiewicz et al., 2009;

2102; 2014; Ward et al., 2013) to explain aspects of community structure.  The theoretical underpinning of size-selected grazing (Armstrong 1994) have similarly been used in many studies (e.g. Lampert, 1997, Kiorboe, 1993; 2008; Schartau et al 2010; Ward et al., 2014; Acevedo-Trejos et al., 2015). The appendix and the insight we develop in the rest of this section are in some sense a synthesis of many prior studies. However here these theories are specifically directed at understanding diversity patterns, something that to our knowledge has not been done before.

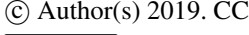



### 4.1 Size Class Diversity

We find that the richness of cell sizes increases with the supply rate of the limiting nutrient (Fig 9). Theoretical predictions and previous model studies suggest that this should be the case when the resource requirements of phytoplankton increase with increasing size (appendix, Armstrong, 1994; Ward et al, 2014; Follows et al 2018). In the nomenclature of resource supply theory (Tilman, 1977), $R^*$ of a phytoplankton type is the minimum resource concentration required to for it to survive at steady state. In the absence of grazing, $R^* = \frac{k_R M}{\mu_{max} - M}$ where $K_R$ is the resource half saturation constant, $\mu_{max}$ is the maximum growth rate and $M$ is a loss rate (see methods). The phytoplankton with the lowest R* will draw the nutrients down to this concentration and exclude all others. In our model, the smallest pico-phytoplankon have the lowest $R^*$ and larger phytoplankton have subsequent higher $R^*$ (Fig 4b). In this formulation, the smallest phytoplankton should out-compete all others. However, when we take grazing by a zooplankton ($Z$) into account, $R^* = \frac{k_R g Z}{\mu_{max} - g Z}$ where $g$ is a per biomass grazing rate. Thus, $R^*$ increases with increased grazing. When the grazing pressure is sufficiently strong on the smallest type, the $R^*$ of the next smallest phytoplankton is reached and the two phytoplankton can co-exist. The smallest size class phytoplankton and its grazer have their biomass capped and any increase in biomass is now due to the next size class (Armstrong 1994). This process continues to more and more size classes as we go from regions of low to high nutrient supply rates (Fig 9a,b,c). In the model, some regions have different limiting nutrients (e.g. iron versus dissolved inorganic nitrogen), so the patterns of size diversity from the total community are more complicated than considering only one nutrient supply rate. However, this process is nicely shown by the number of size classes within the diatom group alone increasing cleanly with the supply of silicic acid (Fig 9d). The fact that each size class is capped by grazing leads the distributions of size classes to be relatively even, especially in the highest nutrient regimes (shown by the Shannon Index, Supplemental text S3, Fig S8).

To explore the importance of size-specific top-down control on diversity suggested by this theoretical construct, we conduct a sensitivity experiment (EXP-1, Table 1), where we allow only one grazer to prey on all phytoplankton. We also do not allow for mixotrophy. We find that only the smallest size class in each functional group survives (Fig 10b, Supplemental Fig S9): the 0.6µm pico-phytoplankton and the 3µm diazotrophs, coccolithophore and diatom. The dinoflagellates do not survive without mixotrophy. The size diversity reduces to one in most regions (Fig 11). This experiment highlights that size diversity (Fig 8b) is controlled not only by the rate of supply of the limiting nutrients, but also by size specific grazing (Armstrong 1994, Poulin and Franks, 2010, Ward et al 2012).

The thermal norm richness of EXP-1 is very similar to the original "default" experiment (Fig 8), and thus richness of this dimension is not (at least greatly) controlled by size specific grazing. Functional groups richness decreases as the dinoflagellates are no longer viable without mixotrophy. All other functional groups survive (Fig 10b, Supplemental Fig S9) and there is coexistence at the functional level; however, the patterns are different to the default experiment. In EXP-1 there are significant changes to the biogeochemistry, including the primary production (lower) and subsequent changes to nutrient supplies. It is these biogeochemical changes that alter the functional richness patterns (discussed more below).





We have used steady state theory to explain the co-existence of size classes. We contend that when looking at annual average richness this theory provides insight even in non-steady state regions such as the highly seasonal latitudes. However, we do acknowledge that the processes are more complex in these regions. For instance, during times of resource saturated conditions (e.g. beginning of the spring blooms), the smallest diatoms, which are the fastest growing phytoplankton, will dominate (Dutkiewicz et al., 2009, see appendix). However, as the grazer of the smallest diatom increases, the phytoplankton net growth rate (growth minuses losses) decreases until the next fastest growing phytoplankton (whose net growth rate is higher since it is not yet under grazer controlled) is able to grow in (Fig 12). Such a progression of size classes of diatoms has been observed using Continuous Plankton Recorder (CPR) data (Barton et al. 2013) and modelled for a coastal system (Terseleer et al, 2014). This process of succession continues until nutrients are drawn down, allowing the pico-phytoplankton and mixotrophs to dominate in this more steady-state low nutrient environment (as suggested by Margalef's mandala, Margalef, 1978). Given that annually there is an optimum condition for each of those size classes, they do all co-exist though at seasonally varying abundances (i.e. they never go extinct locally).

## 4.2 Functional Group Diversity

The size class and functional group classifications are not completely orthogonal as the "pico-phytoplankton" group is entirely composed of the 4 smallest size classes. We therefore use a similar explanation as to why pico-phytoplankton can coexist with the other functional types in low seasonality regions: the pico-phytoplankton low $R*$ allows them to survive ubiquitously and other functional groups can only coexist where (or when) grazing pressures on the pico-phytoplankton and resource supplies are high enough.

We find that for the rest of the functional groups, co-existence is strongly controlled by the differences in their resource requirements and the imbalances in the supply rates of multiple resources (resource supply ratio theory, Tilman 1982, see methods). For instance, slow growing diazotrophs can only co-exist with faster growing other phytoplankton when there is an excess supply of iron and phosphorus delivered relative to the non-diazotrophs N:P and N:Fe demands (Fig 13a,b,c; Ward et al., 2013; Follows et al 2018). In such locations, the non-diazotrophs are nitrogen limited, while the diazotrophs can fix their own nitrogen, and the excess P and Fe not utilized by the non-diazotrophs is available (methods; Fig 13b,c).

Similar arguments explain where non-diatoms can co-exist with the fast-growing diatoms (Fig 13d). In regions where there is excess supply of dissolved inorganic nitrogen, phosphate, and iron relative to the diatom Si:N, Si:Fe, Si:P demands there can be co-existence (Fig 13e,f,g). In these locations (or occasions), diatoms are limited by silicic acid, and any excess N, P and Fe can be used by the other phytoplankton. When the excess supply is significantly high, non-diatoms can dominate. The high silicic acid supply in the Southern Ocean leads to lower diversity as the diatoms win out in all but the low nutrient summer months, when (in this simulation) pico-phytoplankton are the only other functional group to survive. In other seasonal regions, such as the northern North Atlantic (Fig 12) diatoms dominate at the beginning of spring, but coccolithophers can outcompete later in the summer when the diatoms become limited by availability of silicic acid.





The mixotrophs have two sources of resources: inorganic nutrients and other plankton. They are parameterized to photosynthesize slower than other phytoplankton (of the same size, as suggested by observations, Tang 1994; Fig 4a) and graze slower than other grazers (of the same size, Jeong et al., 2010; Supplemental Fig S2). They are advantaged over specialist autotrophs and heterotrophs when competition for both inorganic nutrients and prey is strong and by using both, their $R*$ for each resource is lowered.

To demonstrate that differential nutrient requirements lead to much of the functional group co-existence, we conduct another sensitivity experiment (EXP-2, Table 1) where we force all functional groups to have the same resource requirements (e.g. diatoms do not require silicic acid, diazotrophs cannot fix nitrogen, dinoflagellates cannot graze on other phytoplankton) and C:N:P:Fe ratios are the same for all types. All other growth and grazing parameterizations remain the same as in the default experiment. In this simulation the functional richness reduces dramatically (Fig 11), only pico-phytoplankton and diatoms

survive (Fig 10c, Supplemental Fig S10). The diatoms are the ultimate opportunists (r-strategists) in this model, with the highest growth rate (Fig 4a), and survive when nutrient supplies are high enough. Without any differentiating nutrient requirements relative to the other functional groups, they outcompete them. Pico-phytoplankton (the gleaners, k-strategist) survive in regions of lowest nutrient supply where their low R*, and low grazing allows them to exclude the diatoms. Size class and thermal norm diversity change very little (Fig 11).


### 4.3. Thermal Norm Diversity

We find that thermal norm richness is highest in the regions of the western boundary currents and other regions generally anticipated to have high levels of mixing of different water masses. Clayton et al (2013) identified a link between hot spots of diversity and eddy kinetic energy and the variance in sea surface temperature. Anticipating the role of currents and mixing of

water mass (Clayton et al 2013, Levy et al 2014), in a third sensitivity experiment (EXP-3, Table 1) we do not allow transport of plankton between grid cells, though we do allow diffusion vertically in the water column. Thus, this simulation is a collection of one-dimensional models with regard the plankton. However, nutrients and detrital organic matter are allowed to be transported as in the default experiment. Thermal norm diversity decreases (Fig 11), and there are no longer hot spots. These results echo findings from Levy et al (2014), and clearly show the importance of mixing of water masses for maintaining

thermal norm diversity. When temperature is fluctuating all phytoplankton with different temperature norms can survive together provided their respective temperature optimal occur for long enough (Kremer and Klausmeier, 2017) or there is a constant supply of the types from upstream (Clayton et al., 2013). This is different from resources or grazing control where competition for limited resources is the main process controlling co-existence (or lack thereof), and as such we find greatest effect in EXP-3 on thermal norm diversity.

We find in EXP-3 that the geographical size of almost all habitats (Fig 14, Supplemental Fig S11) is reduced. In the case of thermal norms, lack of transport allows for very little co-existence. For functional diversity, the pattern changes, but the maximum richness remains the same. This suggests that the boundaries of functional groups domains are expanded by transport



(see for instance the decrease expanse of diatoms in the gyres, Supplemental Fig S11 versus S5), but transport per se is not the ultimate controller. Domains for each size class also decrease (Fig 14, Supplemental Fig S9), but most dramatically for the

larger size classes, and the two largest go extinct in this experiment. This suggests that transport also plays a role in maintaining the grazer/phytoplankton links and that for classes with smaller domains and/or very low biomass this becomes more crucial. A few types have an increase in range, or in fact exist in EXP-3 and not in the original experiment (Fig 10d, 14, Supplemental Fig S11). These are almost all the warmest adapted types that in EXP-3 have very small biomass and ranges. Thus, transport can also reduce domains of types with very small potential niches as the constant influx of less fit types from cooler regions is

sufficient to overcome any competitive advantage of the locally superior warm-adapted types.

## 5. Links to Diversity along the Atlantic Meridional Transect

Using the results of this study, we can hypothesize as to why richness of co-existing nano and micro eukaryotes along the AMT (Fig 1a) have the observed patterns. We consider the diversity within the three dimensions along the transect (Fig

15b,c,d). All three dimensions have high diversity along the north edge of the Southern Ocean (labelled A in Fig 15), suggesting that all controls (supply rate of limiting nutrient, imbalance in supply of different nutrients, top-down control, and transport) are at play in setting the maximum richness seen here in both model and observations (Fig 1a,d). Thermal and functional richness decrease southward, leading to the drop in total richness observed poleward. Absolute nutrient supplies are still high enough to maintain size diversity, but the N:Si supply ratios are no longer conducive to maintaining coccolithopores (Fig

13e,f,g) and their diversity decreases as is observed (Fig 1b,e). In this southernmost region there is also no longer the mixing of different water masses between subtropical and Southern Ocean to promote large thermal norm diversity. On the other hand, diatom diversity (due here to size classes) increases (Fig 1b,e), driven by the large gradient in silicic acid supply rate (Fig 9d).

All three dimensions have an even sharper decrease equatorward of the Southern Ocean boundary, leading to much lower total

diversity observed into the South Atlantic subtropical gyre (labelled B in Fig 15). Here the lower absolute nutrient supply leads to reduction in size classes, silicic acid supply rates drop dramatically (Fig 9d) and functional diversity decreases. The lack of mixing of water masses reduces the thermal norm diversity. Nearer the equator (labelled C), both size and functional diversity are high, leading to the observed increase in total diversity. Here an increased supply of nutrients (Fig 9) from equatorial upwelling, including slightly higher Si supply rates are important for allowing additional size classes and diatoms to exist. In

the region of the Canary upwelling region (labelled D), there is an increase in diversity in the model and observations. Here increased size class and thermal norm diversity are responsible, a result of the nutrient-rich upwelled water mixing with surrounding water masses as it is transported offshore (see Clayton et al 2014). The model underestimates this increase since the model's coarse resolution does not capture the meso-scale filaments associated with these upwelling features found in the real ocean.






## 6. Limitations of this Study

This study must be understood within the context of the limitations of the model. Models are by definition simplified constructs that attempt to capture the essence of a real system. The model here has a more complex ecosystem than many other marine models but is still limited in terms of the parameterization choices. For instance, the size dependent grazing assumes a 10 to 1
preference as suggested by observations and used in many other studies (Fenchel 1987; Kiorboe 2008, Ward et al., 2012, Baird et al., 2004). However, there are many examples of grazing that breaks these rules (Jeong et al 2010; Weisse et al 2016; Sommer et al, 2018). The model assumes fixed elemental ratios in the plankton. This too is an oversimplification, and variable ability to store nutrients is an important trait that likely allows for levels of co-existence (Edwards et al 2011) that is not incorporated here. However, the model carries almost 750 unique tracers to account for all the phytoplankton, variable Chl-a
as well as the inorganic and organic pools. To include variable stoichiometry would add over 2000 more tracers that is computationally unfeasible for this study. Each functional group has a different absorption spectrum, though these are modified with size (see Supplemental Fig S1, and text); we recognise that this has a large implication for the pico-phytoplankton whose accessory pigments are quite different. Using a version of this model, but with differing absorption spectra for the pico-phytoplankton, Hickman et al (2010) showed that such difference was responsible for some niche separation, especially
vertically. The results of this study should be interpreted in light of these and other simplifications.

The model considers only three axes of phytoplankton traits. We anticipate that additional axes such as morphology (e.g. shape, spines), motility (e.g. flagella), chains, colony formation, nutrient storage abilities, and symbiosis will each have their own controlling mechanisms. Previous studies have suggested other controllers of phytoplankton distributions when considering other traits, for instance the importance of trade-offs between nutrient acquisition and storage (e.g. Edwards et al
2011) or the effect of symbioses (e.g. Follett et al 2018; Treguer et al 2018).   Here we have specifically designed the model to only consider the three dimensions for simplicity. Including additional trait dimensions will likely lead to alterations to the patterns of diversity, and will be important for follow on studies, especially as our knowledge of the trade-offs of each trait dimension becomes clearer. For instance, that the model underestimate diversity in the subtropical gyres suggests that additional dimensions are likely important in these regions.

Our results are also dependent on the resolution of different axes of trait space. Likely in the real ocean there is a similar (though more complex) coarse resolution of functional groups, but much higher (potentially continuous) resolution of size classes and thermal norms. Total diversity may therefore be influenced more by these two axes than established in this study. This study should be viewed as only a step in the understanding controls of diversity and provide new evidence to explain the 'paradox of the plankton' (Hutchinson, 1961). However, that we can capture the major patterns of the AMT (Fig 1) suggest
that we have included some of the most important mechanisms.



## 7. Discussion

We have used ecological theories and a numerical model to examine the controls on phytoplankton diversity along a number of trait dimensions. We find that each dimension has a different set of controls. Observed "total" diversity is an integrated
function of the richness along each trait dimension and is thus controlled by many different mechanisms. By focusing on the mechanisms, we can understand the patterns of diversity at the fundamental level. Such insight provides us with a perspective to predict changes that might occur in diversity in, for instance, a warming world.

Our results suggest that observed patterns of "total" diversity (or for any grouping of phytoplankton types, such as for nano and micro-eukaryotes along the AMT) are a result of multiple controllers: supply rate of limiting resource, imbalance in supply
of different resources relative to competitor's demands, top-down control, particularly in terms of size-dependent graxing, and transport processes. The importance of both resource supply and resource imbalance (or resource supply ratio) has previously been demonstrated by Cardinale et al (2009) for lake habitats and more recently for other natural phytoplankton assemblages (Lewandowska et al., 2016).

In this study we have synthesised previously known theory and numerical model. The results explain why previous model
results have had sometime contradictory results. In ecosystem models where only considered two dimensions of diversity (functional groups and thermal norms, Barton et al, 2010, Clayton et al 2014) different patterns where obtained relative to a model that only considered size (Ward et al., 2014). For instance, the hotspots of diversity in western boundary currents were not apparent in the study of Ward et al (2014) since thermal norm diversity was not included in that study. Similarly, the lack of high diversity along the edge of the Southern Ocean in Barton et al (2010) that is seen in this study and in the AMT
observations (Fig 1) was due to the lack of size trait dimension in that study. This stresses that "diversity" in models needs to be understood in terms of the traits that are included. This obviously bring up the questions raised in Section 6: What additional patterns will be apparent in models that include additional, or other, trait dimensions. An exciting avenue for future study.

The drivers we found in this study (supply rate of limiting resource, imbalance in supply of different resources relative to the better competitor's demands, size-dependent grazing, and transport processes) have little to do with environmental factors such
as temperature or latitude that have been investigated by correlations to diversity patterns (see e.g. Hillebrand and Azovsky, 2001; Hillebrand, 2004; Irigoien et al. 2004; Smith et al, 2007; Rodriguez-Ramos et al 2015; Powell et al, 2017). Though observational studies have hypothesized a multi-factorial control on diversity in the ocean (e.g. Rodriquez-Ramos et al 2015; Lima-Mendez et al 2015), they were unable to find significant correlations with any combination of environmental factors such as latitude, temperature or biomass, or even nutrient concentrations. Correlating with environmental factors (such as
temperature, latitude) is a logical first step for trying to understand observed patterns of diversity, as these are often the only additional data that is available from a field study. Our study however suggests that to some degree these are the wrong metrics to be considering and are thus unlikely to help disentangle controllers of diversity. For instance, in our study it is mixing of different temperature water masses, potentially hinted at by local temperature variances rather than temperature itself, that is important. Similarly, observations of community structure show little statistical links to nutrient concentrations (e.g. Lima-





Mendez et al., 2015), but would if nutrient supply rates (a harder variable to measure) were used instead (see e.g. Mouriño-Carballido et al. 2016). Diversity controls inferred by correlations with environmental factors or from niche modelling (e.g. Righetti et al 2019, who make use of statistical inferences on species biogeography), likely miss important drivers. For instance, biotic interactions (competition and grazing) and impacts of transport (two mechanisms we have shown to be important) cannot be captured using such statistical techniques.

Biomass and productivity are dictated by the supply rate of the limiting nutrient, and therefore our study found an increase in size diversity with increased productivity and biomass. This compares well to the observations of Marañón et al (2015) and Acevedo-Trejos et al (2018) who found an increase in size classes with higher productivity. However, we caution that it is nutrient supply rate (not productivity) that is the controlling mechanism. However, nutrient supply rate (a bottom up process) cannot alone lead to high size diversity. Top down processes are essential for the buildup of size classes with higher nutrient

supply (see also Poulin and Franks, 2010). Considering only correlations with productivity would lead one to miss this important biotic interaction as a control on diversity. In our model top-down control was size-specific grazing, but similar patterns could be achieved with kill-the-winner type parameterizations (Vallina et al 2014a) or imposing species-specific grazers or viruses.

Though transport of phytoplankton most strongly controls the thermal norm diversity, we did find that it modulates the extent

of the regions for all traits. For instance, diatoms die out in the central subtropical gyres when transport is turned off in EXP-3, and the largest size classes become less competitive without transport (Fig 10d, Supplemental Fig S11). Our explanations of the different controls on the diversity along different trait axes should be understood as focusing on the most important components. The real system has multiple controlling mechanisms working together. This only further emphasizes that correlating diversity with simple environmental factors such as temperature, latitude, productivity, or even nutrient

concentrations will miss that it is a complex set of controllers that are important.

The discussion of marine phytoplankton diversity must also be considered in light of the limited, but also different types, of observational datasets (see review Johnson and Martiny, 2014). Different techniques tend to capture just some aspects of diversity, for instance different axes are distinguished when instruments measure just size (e.g. by Flow Cytometer, LIIST), pigments (e.g. though HPLC), or morphologic and taxon differences (e.g. microscopy). Only recently have studies

incorporated diversity from a genomic perspective (e.g. de Vargas et al, 2015). Genomic diversity tends to capture a much higher diversity than other methods, with a long tail of rare species not captured by other measurement (Busseni, 2018). Thus "diversity" depends on the definition, and/or on the measurement used. Observational datasets are, however, sparse and only capture a small temporal and spatial pattern of biodiversity. The key to having consistent datasets (e.g. Rodriguez-Ramos et al, 2015; Sal et al 2013), or that sampling biases might skew results (Cermeno et al 2013) have only recently become commonly

understood.





## 8. Conclusions

In this study we have disentangled some of the multiple controls on marine phytoplankton diversity. We have shown through theory and a model that diversity within different dimensions of phytoplankton traits are controlled by disparate drivers. The
number of co-existing size classes of phytoplankton is largely controlled by the magnitude of the limiting resource supply rate and the strength of the size-specific top-down processes; functional groups co-existence is partly controlled by the imbalance in the supply rate of different resources relative to competing species' demands; the number of phytoplankton with different thermal optima that can co-exist is strongly controlled by the amount of mixing of different water masses. Transport in general expands the range of phytoplankton habitats and leads to higher local diversity. That each controller affects a different
dimension of diversity is important to explain why diversity patterns in models that include only one or two of the traits will have different results to one that includes all three. Likely including other traits (e.g. morphology, symbioses) controlled by different (as yet not understood) mechanisms will lead to additional components to the patterns of diversity.

This study suggests why there have often been conflicting results in observational studies that have attempted to link diversity to environmental parameters such as temperature or productivity. Such environmental parameters are potentially not the right
metrics to be considering: Even when they do show correlations with diversity, it can sometimes be only because the environmental parameters are also correlated with some of the actual drivers (such as nutrient supply rates), and results will also be specific to the dimensions of diversity measured. Models such as this one can be a good tool to address both consistency and sampling biases, as well as providing insight into controlling mechanisms as we have done here. By understanding the mechanistic controls on diversity we are in a better position to understand how diversity might have been different in the past,
how it changes with interannual variability, and how it might alter in a future ocean.

**Code availability:** The global physical/circulation model (MITgcm) is available at http://mitgcm.org and the ecosystem component is available from git://gud.mit.edu/gud1. Version and modifications used for this study are available at https://doi.org/10.7910/DVN/EOTT9H

**Data availability:** Model output used in this study is available at  https://doi.org/10.7910/DVN/JUQCFG

## Appendix

**Theory:** We consider a system of phytoplankton biomass ($B$) sustained by nutrients ($R$):

$$\frac{dR}{dt} = -\mu_{max}\frac{R}{R+k_R}B + S_R \qquad\qquad Eq\ 1$$

$$\frac{dB}{dt} = \mu_{max}\frac{R}{R+k_R}B - MB \qquad\qquad Eq\ 2$$





Where $\mu_{max}$ is maximum growth rate, $k_R$ is half saturation constant for growth, $S_R$ is supply of resource $R$ and $M$ is the phytoplankton loss term (we will consider different assumptions of M below).

***A1. Steady State:*** Here we synthesis the theoretical underpinning that we have previously presented (Dutkiewicz et al., 2009;

Ward et al., 2013; Ward et al., 2014; Levy et al 2014; Follows et al. 2018). Those studies have in turn been informed from the seminal work of Tilman (1982) and Armstrong (1994).

We assume steady-state and solve the biomass equation (Eq 2):

$$R^* = \frac{k_R M}{\mu_{max} - M} \qquad\qquad Eq\ 3$$

This is the concentration that the phytoplankton will draw the resource down to in steady-state. In a system with $J$

phytoplankton, the one with the lowest $R_j^*$ will draw the nutrients down to this concentration and all others will be excluded.

***A1.1. Grazing allows co-existence:*** If we now consider a system of $J$ phytoplankton ($B_j$) and $K$ zooplankton ($Z_k$), where each phytoplankton has a specific grazer, we can write the loss rate now as $M=m+g_{kj}Z_k$. Here $g_{kj}$ is a grazing rate of zooplankton $k$ on phytoplankton $j$, and $m$ is a linear loss rate (resolving cell death and other losses). In this case:

$$R_j^* = \frac{k_{Rj}\ (m+g_{kj}Z_k)}{\mu_{maxj} - (m+g_{kj}Z_k)} \qquad\qquad Eq\ 4$$

Note that this is not an explicit solution as $Z_k$ is itself a complex function of the parameters. However, this equation can provide us with insight: With higher grazing $R_j^*$ increases. When it becomes large enough it can reach the value of the phytoplankton with the second smallest $R_{j+1}^*$ (assume for now that this second smallest plankton is not grazed) and the two phytoplankton will be able to co-exist. This situation occurs when there is higher resource supply ($S_R$) allowing for a larger biomass of both

phytoplankton and zooplankton. With even higher nutrient supply, similar grazing control of the phytoplankton with the second smallest $R_{j+2}^*$ will allow a third phytoplankton/zooplankton pair to co-exist with the others in the system. This system however does require a separate grazer per phytoplankton, or a strong kill-the-winner parameterization. This theory explains the co-existence of several size classes in the ecosystem model (Fig 8b, Supplemental Fig S5). For more details, see Ward et al (2014) and Follows et al (2018).


***A1.2. Multiple limiting resources allow co-existence:*** If we now consider a system of 2 phytoplankton ($B_j$, where $j$ is 1 or 2) limited by different resources ($R_i$ where $i$ is $A$ or $C$), we suggest that this system can allow for co-existence. To explore when the two types can co-exist we expand Eqs 1 and 2 (where the biomass is in units of element $A$) such that:

$$\frac{dR_A}{dt} = -\mu_{max1}\frac{R_A}{R_A+k_{RA1}}B_1 - \mu_{max2}\frac{R_C}{R_C+k_{RC2}}B_2 + S_{RA} \qquad\qquad Eq\ 5$$

$$\frac{dR_C}{dt} = -\mu_{max1}\frac{R_A}{R_A+k_{RA1}}\Upsilon_{AC1}B_1 - \mu_{max2}\frac{R_C}{R_C+k_{RC2}}\Upsilon_{AC2}B_2 + S_{RC} \qquad\qquad Eq\ 6$$

$$\frac{dB_1}{dt} = \mu_{max1}\frac{R_A}{R_A+k_{RA1}}B_1 - M_1 B_1 \qquad\qquad Eq\ 7$$




$$\frac{dB_2}{dt} = \mu_{max2} \frac{R_C}{R_C + k_{RC2}} B_2 - M_2 B_2 \qquad Eq\ 8$$

Where ($\Upsilon_{AC1}$) is stoichiometric ratio requirements of $B_1$ for element $A$ and $C$. $S_{RA}$ and $S_{RC}$ are the supply rate of nutrient A and C respectively. If one of the phytoplankton ($B_1$) has a much higher growth rate than the other ($B_2$) it will be a better competitor for both resources ($A$ and $C$). We find, solving the above equations in steady state that co-existence is possible if:

$$\frac{S_{RA}}{S_{RC}} > \Upsilon_{AC1} \qquad Eq9$$

There must be excess supply of the resource limiting the slower growing phytoplankton relative the stoichiometric demands of the faster growing phytoplankton.

For the case of a Fe limited diazotroph (which can fix their own nitrogen) and a faster growing DIN limited non-diazotroph, co-existence occurs when $\frac{S_{Fe}}{S_N} > \Upsilon_{NFe1}$, where $\Upsilon_{NFe1}$ is the stoichiometric demands of the non-diazotroph. We can write a similar in-equality for any other nutrient limiting the diazotrophs (e.g. P), and find that diazotrophs survive where both $S_{Fe}$ and $S_P$ are supplied in excess of the non-diazotroph requirements (Fig 13b,c). See Ward et al (2013), and Follows et al (2018) for more details.

Similarly, the equations in steady state suggest that for slower growing non-diatoms to co-exist with the fast growing diatoms, the diatoms must be silicic acid limited. In a situation where the non-diatoms are DIN limited, then co-existence occurs if $\frac{S_N}{S_{Si}} > \Upsilon_{SiN1}$ where $\Upsilon_{SiN1}$ is the stoichiometric demands of the diatom. Again, similar in-equalities are applicable if other nutrients limit the non-diatoms (e.g. P, Fe) and we find that non-diatoms can exist where DIN, N and P are supplied in excess of the diatoms requirements (Fig 13e,f,g).

***A1.3. Physical Transport can allow co-existence:*** As discussed in Levy et al (2014), physical transport can also modify $R^*$. Here were recognize that Eq 2 should be expanded for a moving ocean to:

$$\frac{dB}{dt} = \mu_{max} \frac{R}{R + k_R} B - MB + TB + VB \qquad Eq\ 10$$

Where T represents the per unit biomass advection of plankton, $T = -\frac{1}{B} \nabla . \vec{u} B$, where $\vec{u}$ is the local three dimensional velocity vector, and $V$ represents per unit biomass vertical mixing, $V = \frac{1}{B} \frac{\partial}{\partial z} (K \frac{\partial B}{\partial z})$, where K is the vertical mixing coefficient and $z$ indicates the vertical dimension. With these additions,

$$R^* = \frac{k_R(M-T-V)}{\mu_{max} - (M-T-V)} \qquad Eq\ 11$$

Thus $T$ and $V$ provide additional means for phytoplankton to have similar $R^*$. If a phytoplankton type is less competitive at a location, it can still have a similar $R^*$ to a locally better adapted type if there is a steady influx of it from an upstream location. We clearly see this effect in the (generally) expanded biogeography of phytoplankton with advection relative to the experiment without advection (Fig 15, Supplemental Fig 11).



**A2. Non-steady state:** In a previous study (Dutkiewicz et al., 2009) we found that this steady state theory was applicable in a model in the subtropics and in the summer months in some of the high latitude regions. We contend that when looking at annual co-existence this theoretical understanding still provides insight even in non-steady state regions such as the highly

seasonal high latitudes (as was done in Ward et al 2014). However, we do acknowledge that the processes are more complex in these regions. Such regions generally have a succession of dominance of different types. As long as there is a long enough period of favourable conditions for each type, the phytoplankton can co-exist, though with seasonally varying biomass. We explain the succession by considering Eq. 2 in a non-steady state:

$$\frac{1}{B}\frac{dB}{dt} = \mu_{max}\frac{R}{R+k_R} - M \qquad\qquad Eq\ 12$$

Such that the biomass normalized tendency term is dictated by the net growth rate: $(\mu_{max}\frac{R}{R+k_R} - M)$. At any moment (or with a short lag) the phytoplankton with the largest net growth rate can dominate temporally.

*A2.1. Spring Bloom:* As suggested in Dutkiewicz et al (2009), the fastest growing phytoplankton will dominate at the beginning of the spring bloom when the nutrients are plentiful $\frac{R}{R+k_R} \sim 1$, and grazing is small, such that Eq 12 reduces to:

$$\frac{1}{B}\frac{dB}{dt} = \mu_{max} \qquad\qquad Eq\ 13$$

That is the phytoplankton with the largest $\mu_{max}$ will dominate. In the model here, this is the smallest diatoms.

*A2.2. Grazing allows co-existence:* If we now consider two phytoplankton ($B_1$, $B_2$) both limited by the same nutrient, $R$, and each having its own specific grazer ($Z_1$, $Z_2$), so that $M=m+g_{kj}Z_k$. If we assume $\mu_{max1} > \mu_{max2}$, then $B_1$ will dominate when

there is no grazer control. However, when $Z_1$ is large enough, and $Z_2$ is small or negligible, it is possible for

$$\mu_{max1}\frac{R}{R+k_{R1}} - m - g_{11}Z_1 < \mu_{max2}\frac{R}{R+k_{R2}} - m \qquad\qquad Eq\ 14$$

In this case $B_2$ can grow in and potentially dominate the system temporally. Similarly, as grazing control limits $B_2$, a third species with slower growth but also lower grazing might be able to follow on the succession. This is shown in the model for a location in the North Atlantic with a succession of diatoms of increasing size in the spring bloom period.


*A2.3. Multiple limiting resources allow co-existence:* We can also consider equations 7 and 8 (two phytoplankton types limited by different nutrient) in a non-steady state case. If $B_1$ is the faster growing species, it may still be outcompeted (at least temporally) by the slower growing species if

$$\mu_{max1}\frac{R_A}{R_A+k_{RA1}} < \mu_{max2}\frac{R_C}{R_C+k_{RC2}} \qquad\qquad Eq\ 15$$

That is, $B_2$ can succeed $B_1$ if the nutrient limitation of $B_1$ becomes severe enough that its net growth drops lower than its competitor which is limited (less) by a different nutrient. An example is a strongly silicic acid limited diatom later in the seasonal progression succeeded by a nitrate limited coccolithophores, as in the model example (Fig 12). Provided each type





has sufficiently long in favourable conditions each year, it will continue to co-exist at any location though at lower abundances for part of the year.


***A2.4. Physical transport allows co-existence:*** We can use the biomass normalized tendency formulation to consider the circumstances were physical transport has an impact (see Eq 10):

$$\frac{1}{B}\frac{dB}{dt} = \mu_{max}\frac{R}{R+k_R} - M + T + V \qquad\qquad\qquad \textbf{\textit{Eq 16}}$$

Temporarily a phytoplankton type might have the fastest tendency if *T* or *V* are particularly strong (i.e. there is strong supply

of that type to the location through advection or mixing). Such circumstances may occur in highly energetic regions where there is a constant advected supply of different types (e.g. a fast moving Western Boundary Current). A highly varying set of environmental conditions will also help in this situation. For instance if $\mu_{max}$ is assumed to have temperature mediated component (as in the numerical model, Supplemental text S1.4, Eq. S1.4), then many different types would have temporarily "best" environment. However, these beneficial conditions may not occur often enough or long enough to maintain co-existence

without the constant supply of new population. This is the situation in the hot spots of diversity seen in the default experiment, but which disappear in the experiment with no advection (Fig 11). See more discussion in Clayton et al., (2014). We note that the hotspots do not appear in either the size class or functional group richness, suggesting that the temporal "best" environment can be provided by varying temperatures, but no such temporary optimal situation occur in these circumstances for the other dimensional controls


**Supplemental link**

https://doi.org/10.7910/DVN/KXABE6

**Author Contribution**

S.D conceived the experimental design, conducted the biogeochemical/ecosystem/optical model simulations, and performed most of the analysis. O.J. was responsible for the numerical code, with input from AH on the phytoplankton absorption (Supplemental Fig 1). CP provided the AMT observational data (Fig 1a,b,c). BW and MJ provided input on the theoretical interpretations. DT provided input on the grazing parameterizations and the data for Supplemental Fig S2. S.D led the writing

with input from all authors.



**Competing interests**

The authors declare that they have no conflict of interest.


**Acknowledgements**

SD, MJF, OJ received funding from NASA (grants NNX16AR47G, 80NSSC17K0561). This work was also supported by the Simons Collaboration on Computational Biogeochemical Modeling of Marine Ecosystems/CBIOMES) (Grant Id: 549931).

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






|  | **EXP-0** | **EXP-1** | **EXP-2** | **EXP-3** |
|---|---|---|---|---|
| Number grazers | 16 | 1 | 16 | 16 |
| Nutrient requirements of functional groups | Differing | Differing | Same | Differing |
| Horizontal transport of plankton | Yes | Yes | Yes | No |

*Table of sensitivity experiments.*





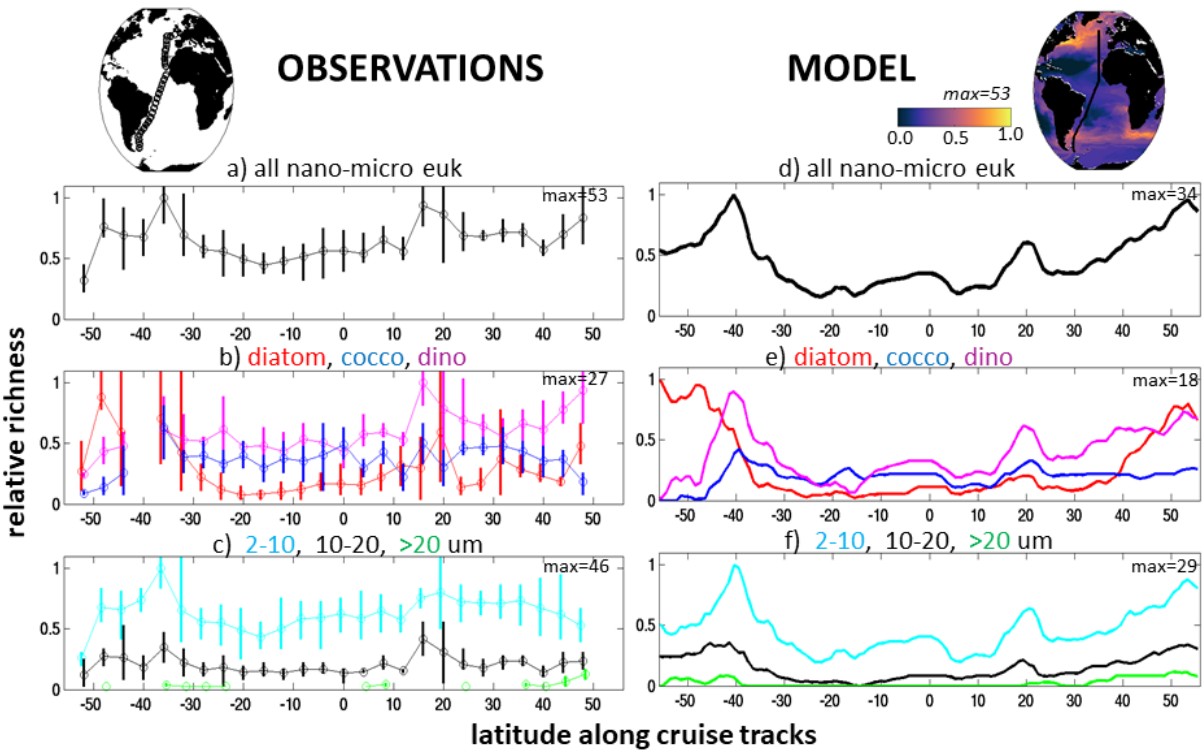

**Fig 1: Nano- and micro-eukaryote normalized richness in the Atlantic**. Left: richness (number of co-existing species)

normalized to the maximum along the Atlantic Meridional Transects (AMT) 1,2,3,4 for microscopy counts (see methods).

Right: normalized annual mean richness from model. (a),(d) all diatoms, coccolithophores and dinoflagellates together; (b),(e)

each functional groups separately (red: diatoms, dark blue: coccolithophores, purple: dinoflagellates); (c),(f) 3 size classes

(light blue: 2-10μm, black: 10-20μm, green: <20μm). In left panels, circles are mean of four transects (2 in May, 2 in

September) within 4º latitude bins, the vertical lines indicate the range within each bin. The maximum number used to

normalise the plots are provided in each panel. Model pico-phytoplankton and diazotrophs are not included in the model

analysis as they were not included in the observations. Maps show the cruise track of the AMTs, and for the model includes

the annual mean normalized richness of the diatoms, coccolithophores and dinoflagellates together.





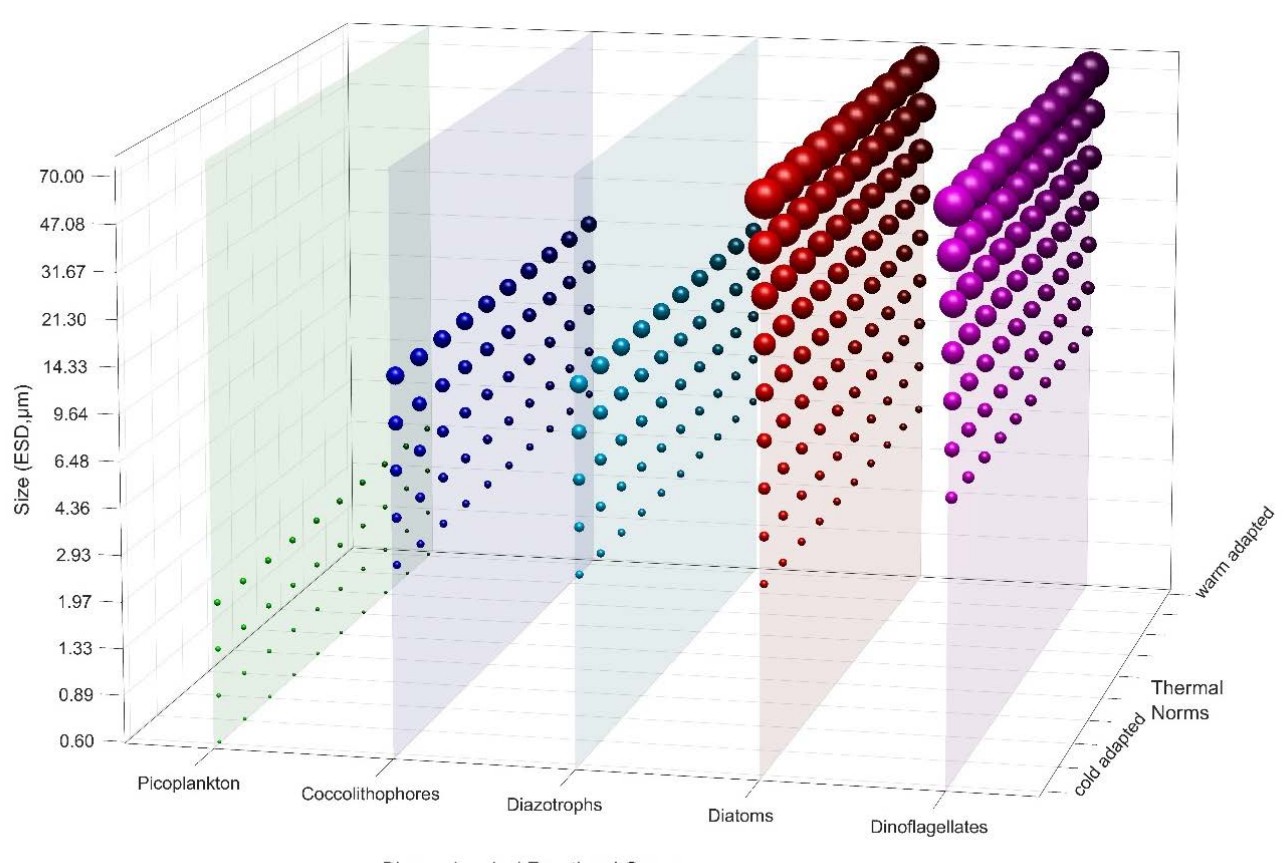


**Figure 2: Schematic of the three dimensions of trait space**: size classes, biogeochemical functional groups and thermal norms. In the actual model there are 16 size classes, 5 functional groups and 10 thermal norms. In all there are 350 individual phytoplankton types. However, the 3 largest size classes go extinct, and as such here we show (here and in other figures) only 13 size classes.



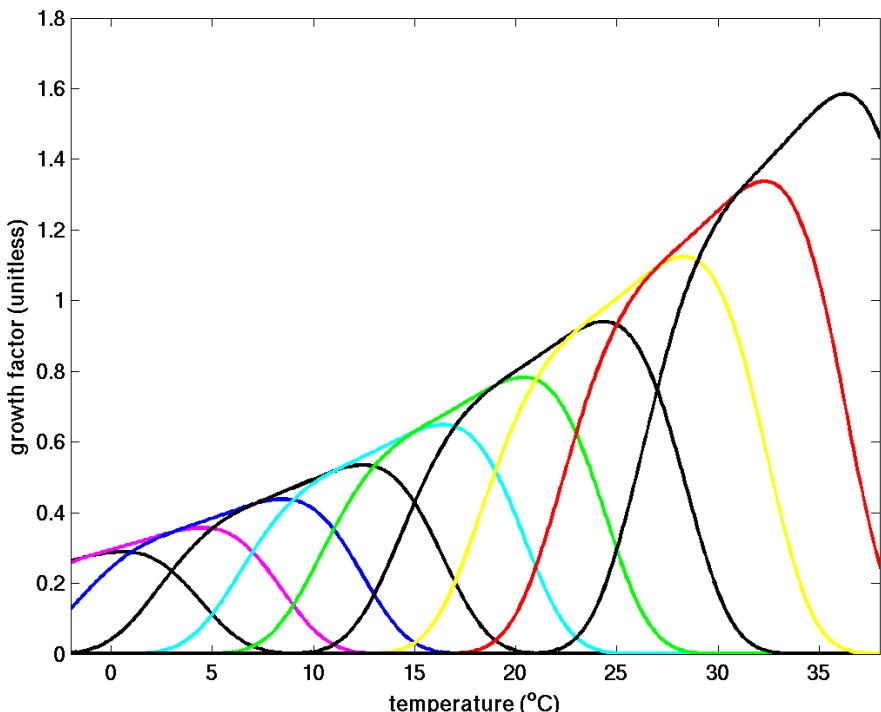


**Figure 3: Growth as a function of temperature.** Shown are the 10 thermal norms (unitless), each with a different colour. The function used here is from Dutkiewicz et al (2015b) and is discussed further in Supplemental material.



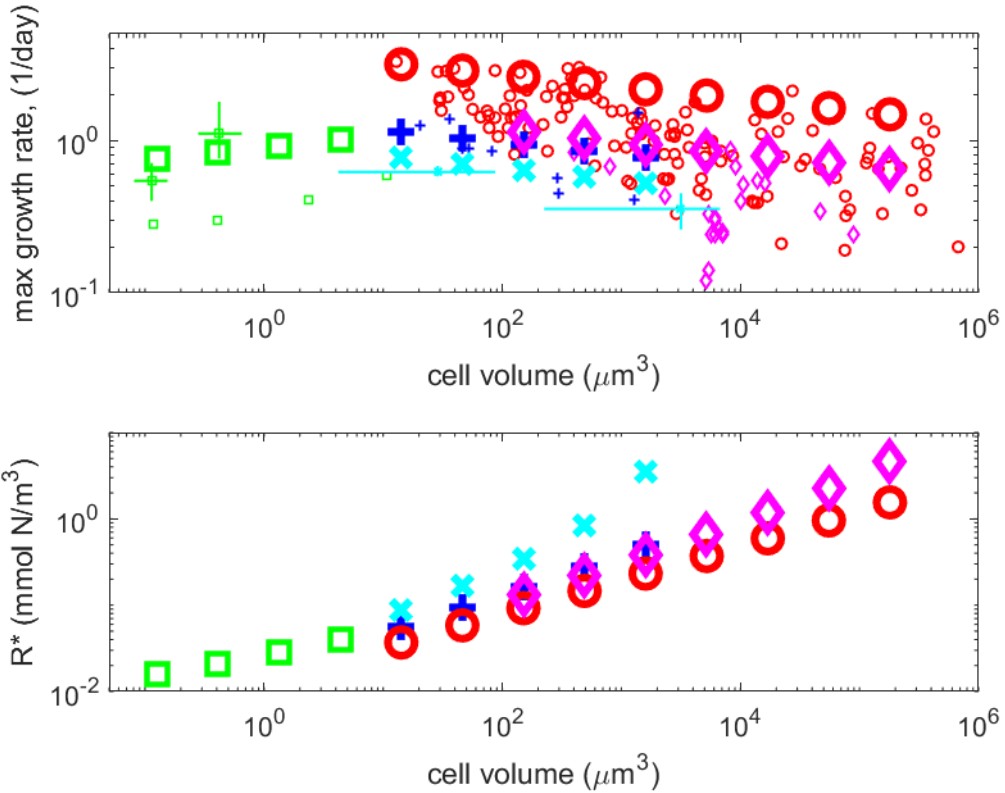

**Figure 4: Model Parameters guide by laboratory studies.** Phytoplankton maximum growth rate (top) and R\* (bottom) as a function of cell size. In (a) small symbols indicate laboratory studies normalized to 20° C, large symbols indicate the model size/functional groups. Colour of symbols denotes different functional groups: red circle=diatoms; purple diamond=mixotrophic dinoflagellates; dark blue plus=coccolithophores; light blue cross=diazotrophs; green square=pico-phytoplankton. In (b), $R^* = \frac{k_R M}{\mu_{max} - M}$, where $M$=0.5 1/d (see appendix). Data compilations of concurrent size and growth in (a) are from Tang (1995); Maranon et al. (2013); Sarthou et al (2005); Buitenhuis et al (2008). Additional data are derived from separate measurement of size and growth: These are shown as light lines centered at the mean and arms covering range. These are for the pico-prokaryotes (green) *Prochlorococcus* and *Synecochoccus* (Morel et al., 1993, Johnson et al. 2006, Christaki et al. 1999, Moore et al. 1998, Agawin and Agustí 1997) and the diazotrophs (light blue) *Crocosphera* and *Trichodesmium* (Garcia and Hutchins, 2014; Webb et al, 2009; Wilson et al, 2017; Bergman et al, 2013; Boatman et al 2017; Beithbarth et al, 2008; Hutchins et al 2007; Kranz et al., 2010; Shi et al, 2012).





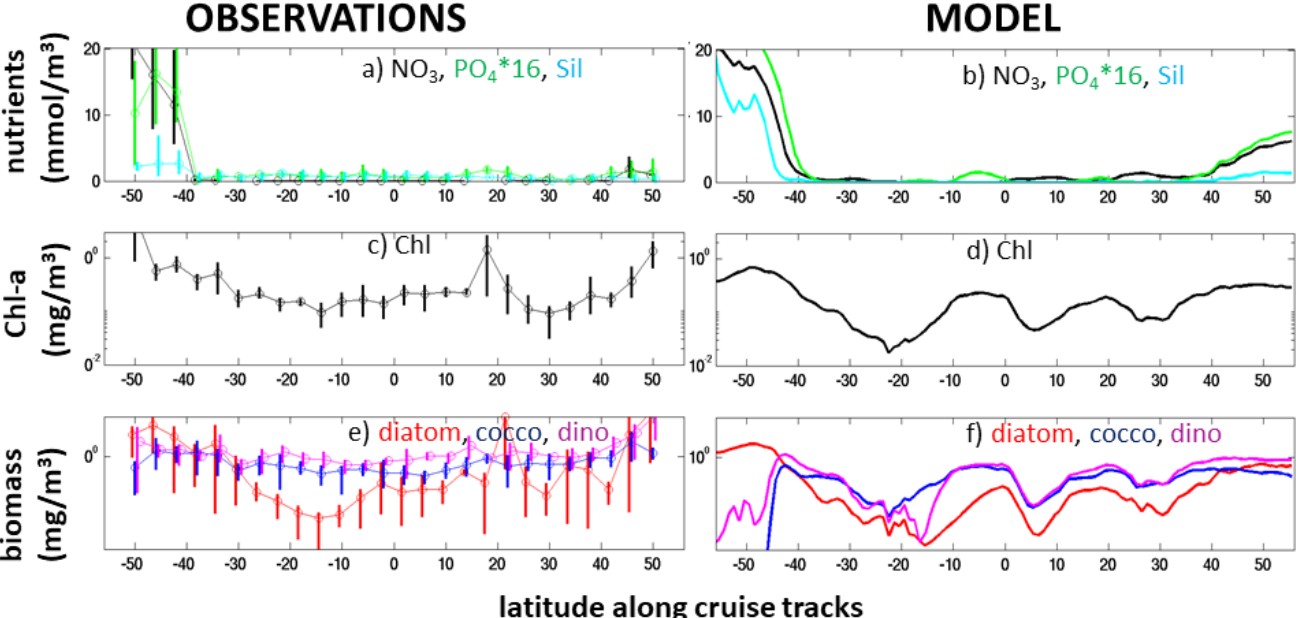

**Figure 5: Observations and model output along the Atlantic Meridional Transect (AMT).** (a), (b) nutrients

(black=nitrate, mmolN/m³; green=phosphate, 16xmmolP/m³; light blue=silicic acid, mmolSi/m³); (c), (d) Chl-a (mg Chl/m³);

(e), (f) phytoplankton biomass (mg C/m³; red=diatoms; blue=coccolithophores; purple=dinoflagellates). Observations (left

panels) are mean (circles) for the 4 AMT cruises (2 in May, 2 in September, see methods) in 4° bins, the vertical lines show

the range within each bin. Model results are annual mean along the AMT cruise track.

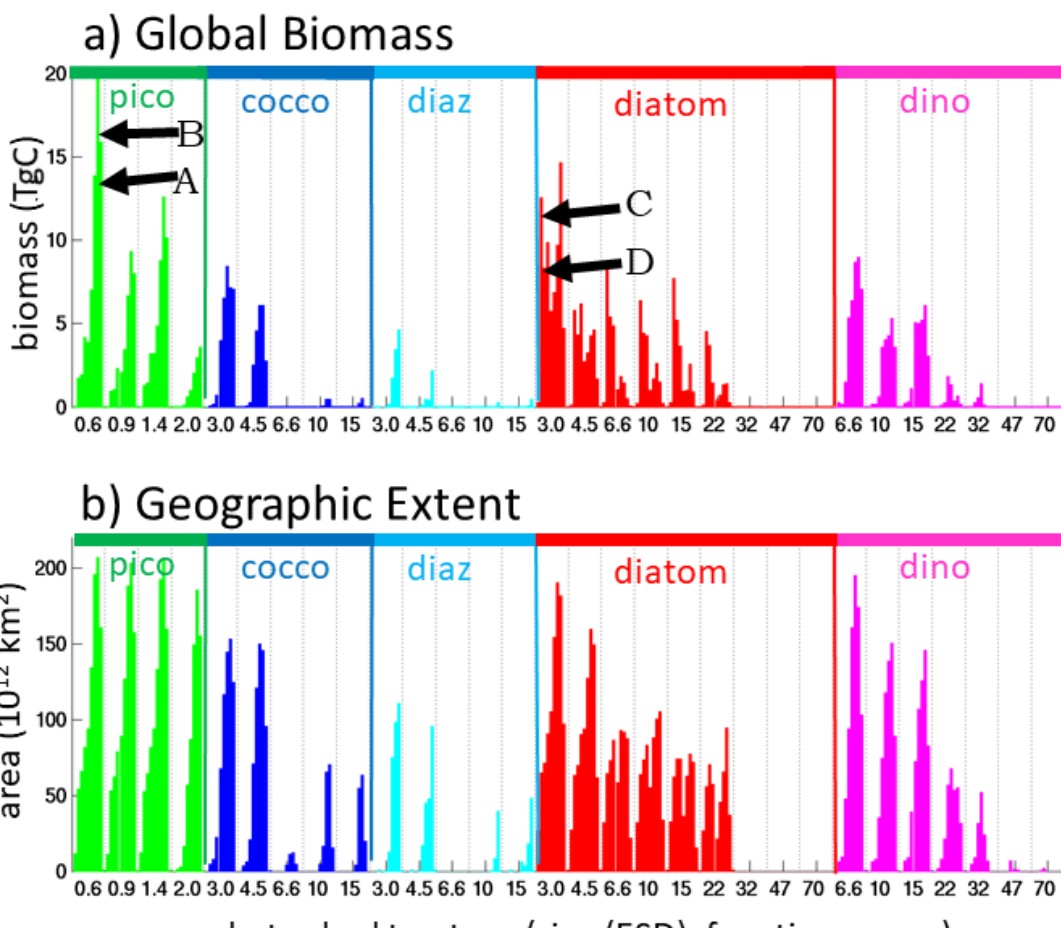

**Figure 6: Model phytoplankton types biomass and range**. (a) Global integrated biomass (TgC); (b) Areal extent of the type ($10^{12}$ km$^2$). Types are arranged by functional group as indicated by the colour bar and labels at the top of the graph, by size classes (equivalent spherical diameter, ESD) as labelled below the graph, and thermal norms from cold adapted to warm adapted from left to right in between vertical dotted lines. The text (A,B,C,D) in panel (a) refers to representative types whose distributions are shown in Supplemental Fig 4.





**Figure 7. Comparison to Observations**. (a) Sizes Classes: Chl-a concentration (mg Chl/m3) in pico (<2um), nano (2-20um) and micro (>20um) phytoplankton from (left) a satellite derived estimate (Ward, 2015) and (right) default model (0-50m); and (b) Functional groups (top) default model (0-50m) and (bottom) data compilation (MAREDAT, Buitenhuis et al 2013) in carbon biomass (mgC/m$^3$). Note the difference in units for (a) and (b) which are chosen to match the appropriate observations. For the MAREDAT databases: pico-phytoplankton (Buitenhuis et al 2012); coccolithophores (O'Brien et al 2013); diazotrophs (Lou et al 2012); diatoms (LeBlanc et al 2012). There was no MAREDAT dataset for dinoflagellates.





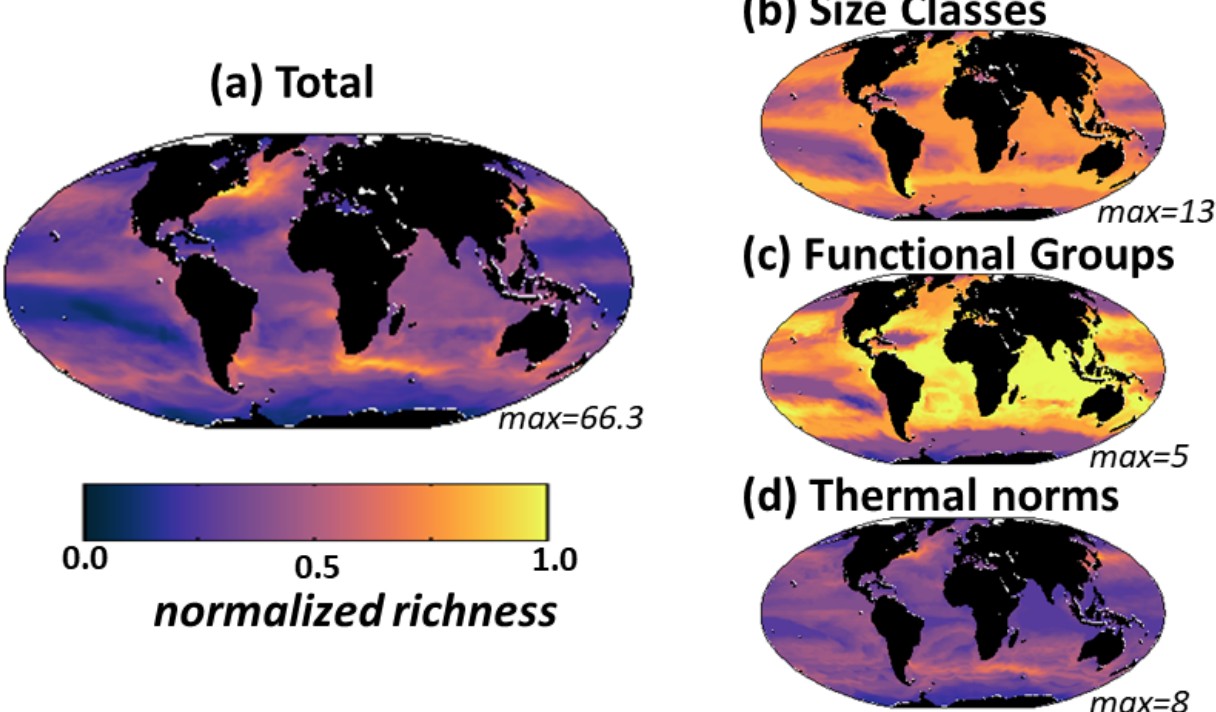

**Figure 8: Model diversity measured as annual mean normalized richness in the surface layer.** Normalized is by the maximum value for that plot (value noted bottom right of each panel). (a) total richness determined by number of individual phytoplankton types that co-exist at any location; (b) size class richness determined by number of co-existing size classes; (c) functional richness determined by number of co-existing biogeochemical functional groups; (d) thermal richness determined by number of co-existing temperature norms. Total richness (a) is a (complex) multiplicative function of the three sub-richness categories (b-d).



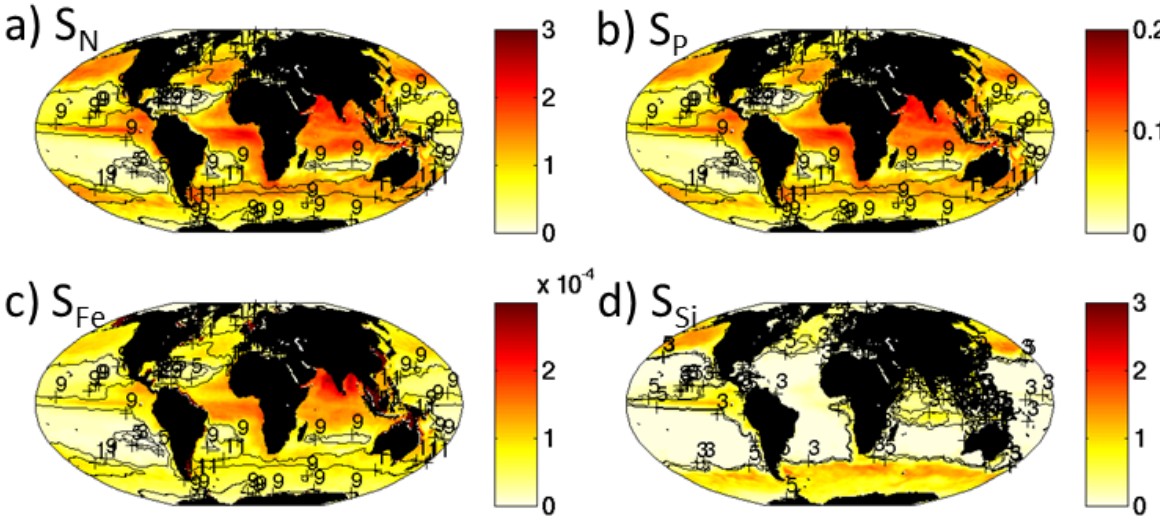

**Figure 9: Model rate of supply of nutrients into top 50m.** (a) Dissolved inorganic nitrogen (mol N/m²/y); (b) Phosphate (mol P/m²/y); (c) Iron (mol Fe/m²/y); (d) Silicic acid (mol Si/m²/y). All transport, diffusion and remineralization terms are included, and for iron also dust supply. In a-c, contours are size class richness from total phytoplankton community (Fig. 4b), and in d contour is for size classes within diatom functional group alone. Since there are multiple limiting nutrients (especially for the non-diatoms), patterns of size diversity shown in a,b,c do not exactly match any single nutrient supply rate. However,

the link between size classes of diatoms and silicic acid supply are clear in d.





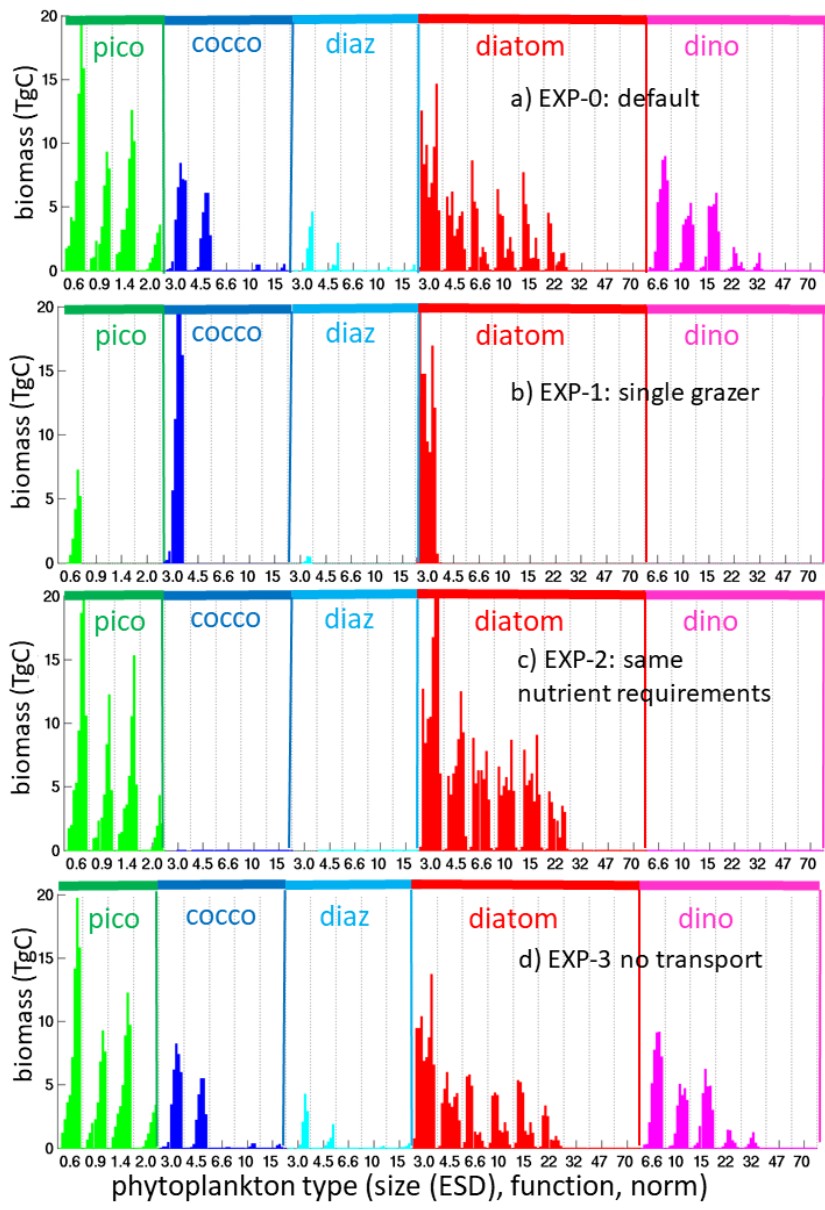

**Figure 10: Sensitivity Experiments, phytoplankton global biomass**. Global integrated biomass (TgC) for (a) default experiment (identical to Fig 3a); (b) EXP-1 (experiment with single generalist grazer); (c) EXP-2 (experiment where all phytoplankton have same nutrient requirements); (d) EXP-3 (experiment where phytoplankton are not transported). Types are arranged by functional group as indicated by the colour bar and labels at the top of the graph, by size classes (equivalent spherical diameter, ESD) as labelled below the graph, and thermal norms from cold adapted to warm adapted from left to right in between vertical dotted lines.




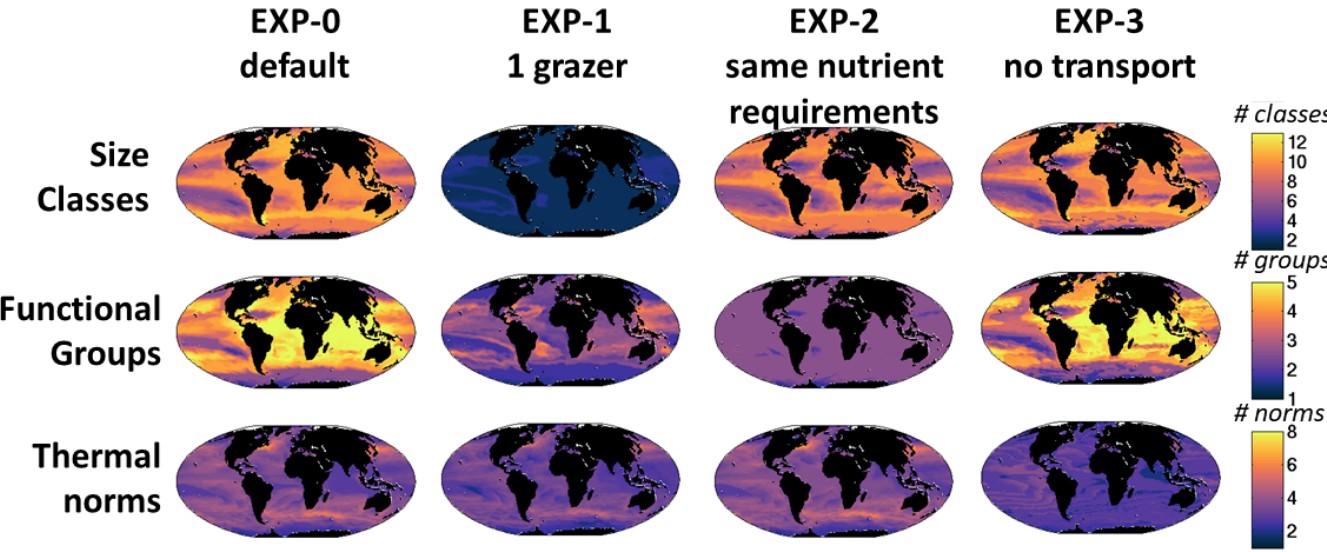


**Figure 11: Sensitivity simulations, model annual mean richness for trait dimensions**. EXP-1 has no size-dependent loss rates (i.e. only one grazer); EXP-2 has no nutrient requirement differences between functional groups; EXP-3 has no transport of the plankton (all nutrients and non-living organic pools are still transported). Top row: size class richness determined by number of co-existing size classes; Middle row: functional richness determined by number of co-existing biogeochemical

functional groups; Bottom row: thermal richness determined by number of co-existing temperature norms. The left most column are the same output as shown in Fig 9b,c,d for the original ("default") experiment.





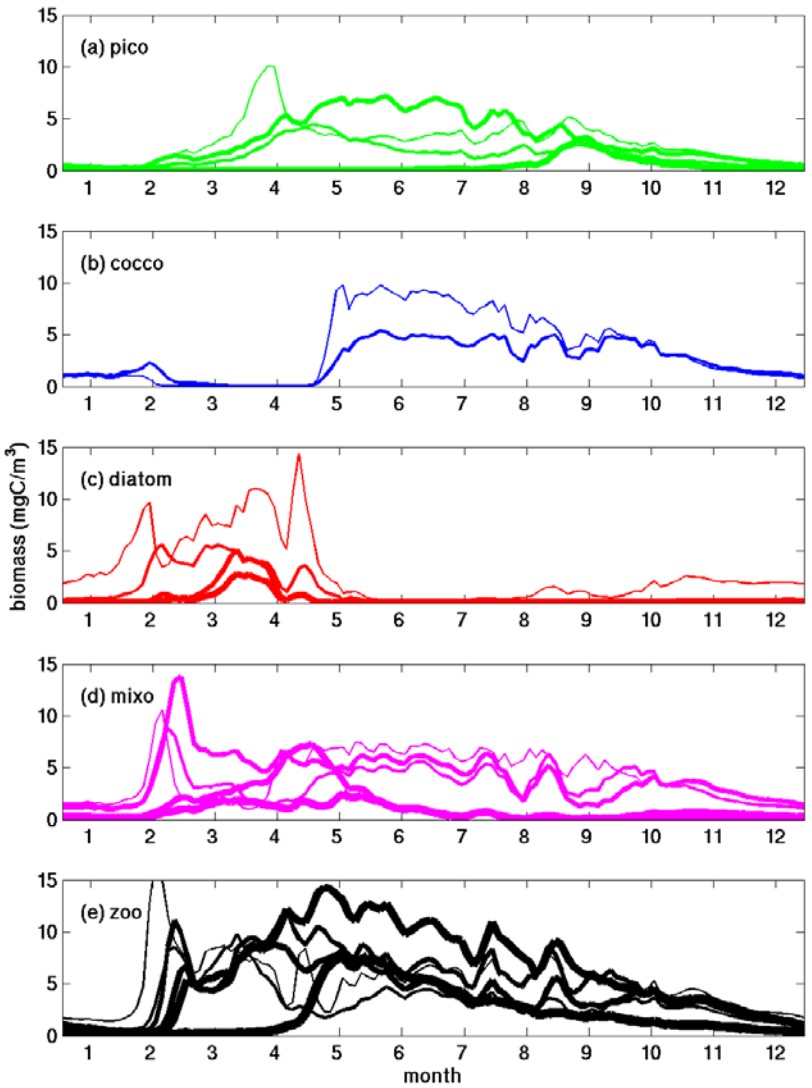

**Figure 12: Default model timeseries in the North Atlantic (20ºW,45ºN).** Carbon biomass (mg/m3) of (a) pico-phytoplankton functional group binned by size class; (b) coccolithophores binned by size class; (c) diatoms binned by size class; (d) mixotrophic dinoflagellates binned by size class; (e) zooplankton by size class. Diazotrophs do not survive at this location. Thickness of lines are based from the smallest to the largest size in each functional group (i.e. thinnest line is for 0.6um for picophytoplankton, 3um for diatoms etc), except for zooplankton where the thickness of line is linked to the preferential diatom prey size (i.e. 30um ESD zooplantkon for the thinnest line), to show the zooplankton-diatom interactions.





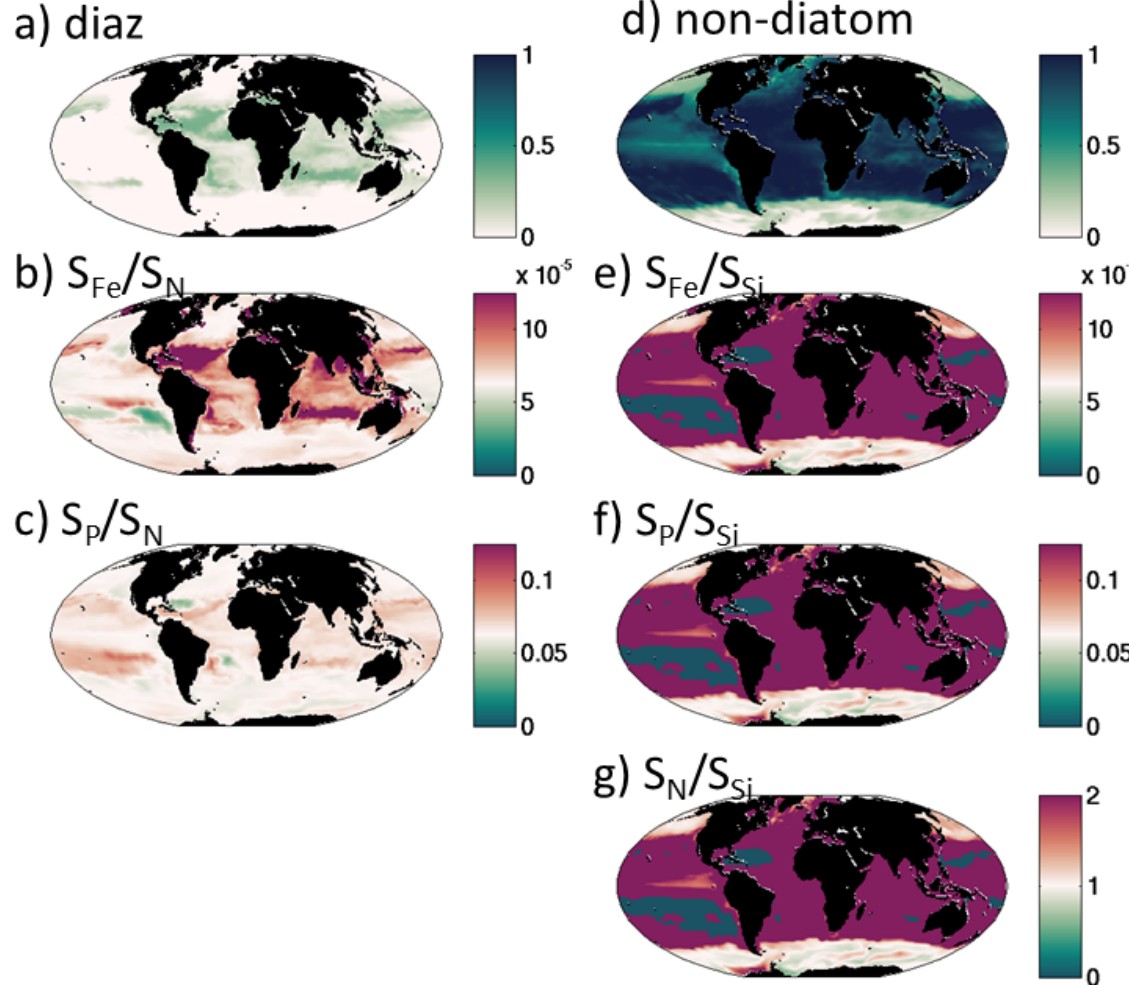

**Figure 13: Co-existence of functional types defined by imbalance of different nutrient supply rates.** Left column depicts controls on diazotroph distribution: (a) fraction of total biomass made up of diazotrophs; (b) ratio of iron to DIN supply rates (see Fig 10); (c) ratio of phosphate to DIN supply rate. Colour scale is chosen such that purple indicates supply rate ratios in excess of the non-diazotroph Fe:N and P:N requirements. Right panel for co-existence of diatoms and non-diatoms: (d) fraction of biomass made up of non-diatoms; (e) ratio of iron to silicic acid supply rates; (f) ratio of phosphate to silicic acid supply rate, (g) ratio of DIN to silicic acid supply rates. Colour scale is chosen such that purple indicates supply rate ratios in excess of the diatom Fe:Si, P:Si, and N:Si requirements.


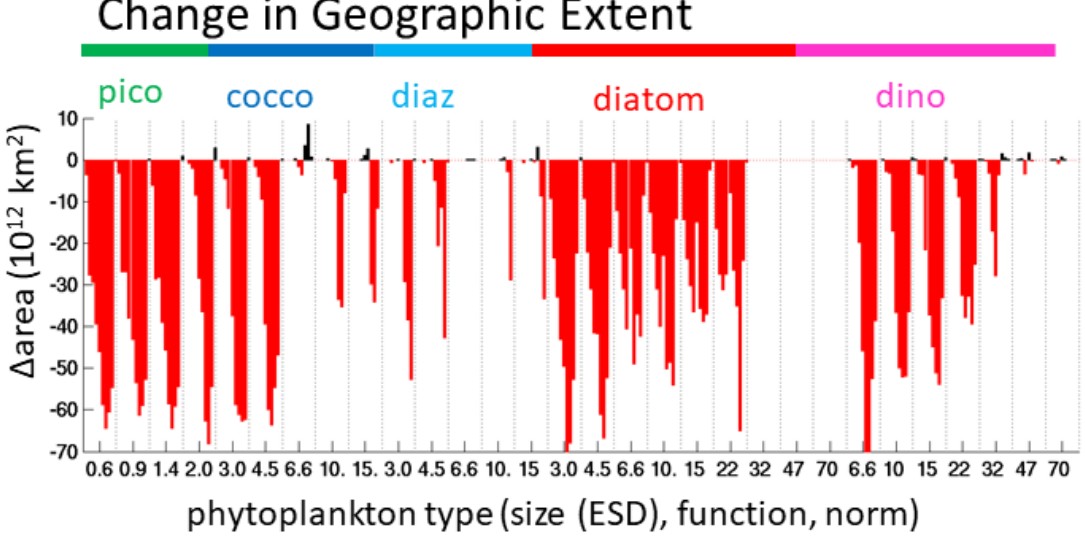

**Figure 14: Difference in phytoplankton range geographic extent**. Change in areal extent of the type ($10^{12}$ km$^2$) between EXP-0 and EXP-3 (no horizontal transport of phytoplankton). Negative (red) indicates a decrease in the geographic domain of the phytoplankton type. Types are arranged by functional group as indicated by the colour bar and labels at the top of the graph, by size classes (equivalent spherical diameter, ESD) as labelled below the graph, and thermal norms from cold adapted to warm adapted from left to right in between each vertical dotted line. Differences are relative to those shown in Fig 6b.
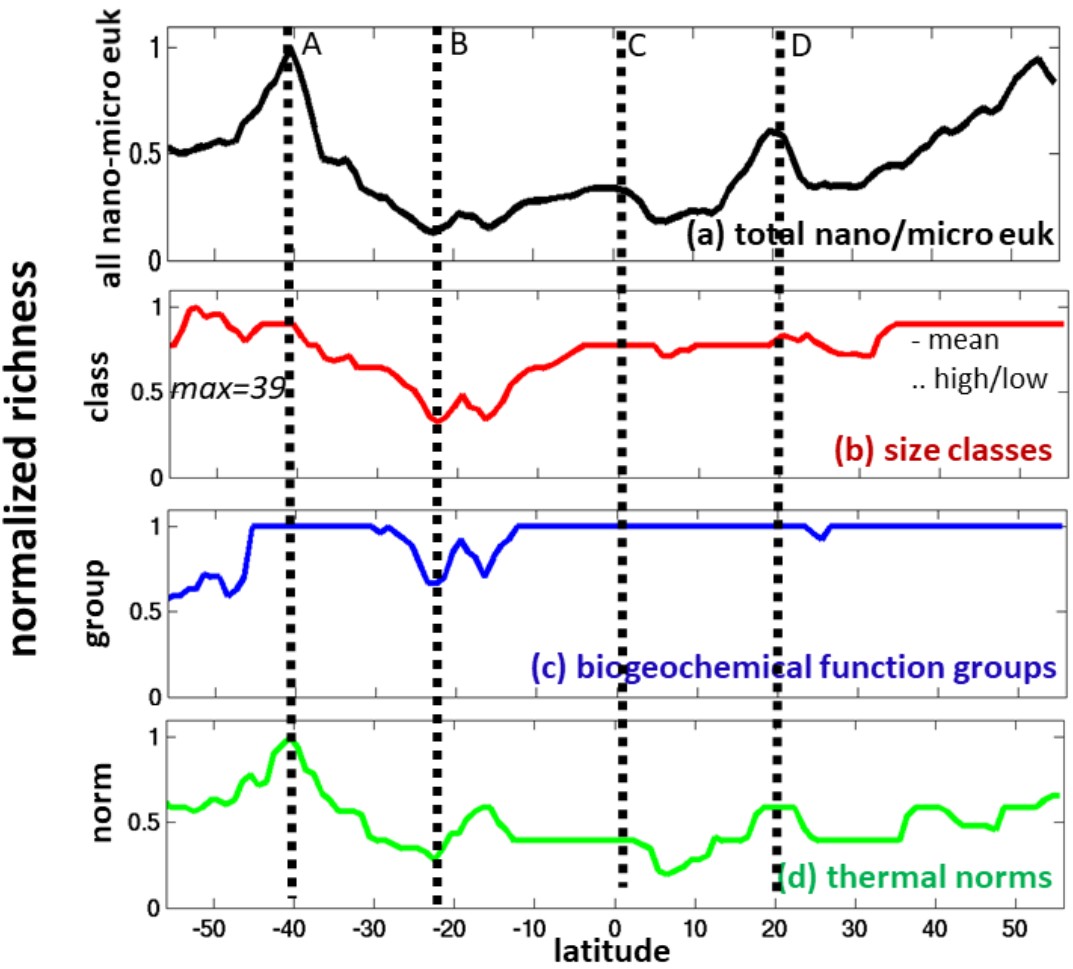

1075

**Figure 15: Modelled nano-and micro eukaryote normalized richness along Atlantic transect, total and for each dimension.** Annual mean richness normalized to the maximum in a transect similar to AMT for (a) all diatoms, coccolithophores and dinoflagellates (maximum of 34), this panel is the same as Fig 1b; (b) size classes (maximum of 9); (c) biogeochemical functional groups (maximum of 3); (d) thermal norms (maximum of 8). Note that pico-phytoplankton and diazotrophs are not included in this analysis as they were not part of the observations. Dashed lines and text (A,B,C,D) are used to locate regions discussed in the text.

1085