# Peer review of "Dimensions of Marine Phytoplankton Diversity"

_Biogeosciences, 2019_

## Referee Comment (RC1) · Anonymous Referee #1 · 20 Oct 2019

The MS by Dutkiewicz et al. presents a detailed account on the drivers of the marine phytoplankton diversity in a numerical model.

The effort is of great interest since the model considers all the processes that are considered relevant when using a trait-based framework. It is also of interest since the impact the main traits and processes are discussed separately, using a set of well-defined sensitivity experiments. Finally, it focusses the discussion on the immediate implications for the interpretation of real data. In particular it points out that the selection of the environmental variables that are used as explanatory variables in statistical analyses has to be coherent with hypotheses drawn from the current theories. Dramatically, if one uses a trait-based framework the outcome is that most of the variables are hard to be constrained quantitatively (eg, the nutrient fluxes).

Indeed, the actual phytoplankton richness is possibly orders of magnitudes higher than the one emerging from this model exercise. This limitation is possibly due to the strong limitations of the "classical" trait-based approach. In addition, in this exercise there are no significant conceptual novelties on specific processes. Nevertheless, as stated by the authors, a general synthesis of the lessons learnt using this framework is going to be very useful for future studies and, with given its pedagogic clarity, for students and young researchers.

I thus recommend it for publication after addressing some very minor points.

A check is required for all the citations (missing parenthesis or points).

Specific comments:

Introduction.

The last sentence is generally correct for the whole diversity but in most cases studies focus on single groups. Is it still true?

Introduction:

A statement on the different definitions of "diversity" is missing. A general issue with the literature on plankton is the lack of discussion about the importance and the technical and ecological implications of the choice of the metrics for diversity.

The study by Lima-Mendez is not on diversity but on interactions. Their conclusions that biotic interactions are more important than environmental factors in setting the community network derive from the analysis of a dataset that contains much more "species" than this model. I think it is just not possible to compare the two approaches with the current state of understanding. In addition, the model has no interactions except for grazing. Thus citing it it is useful especially to discuss how these results represent a challenge for the current modelling approaches.

L45 "there is evidence suggest"?

Section 2.

What is the definition of richness used for the AMT data? What is the reason for not using a rarefaction of the data prior to define richness? The issue should be discussed shortly, also considering the method used here (L94-95).

The model resolution is very low for the current standards for the ocean physics. Presumably, the computational requirements to run the biogeochemical model are such that using a higher resolution was too demanding. Nevertheless, in discussing the limits of the study the lack of mesoscale and submesoscale processes should be mentioned.

L146 Missing the verb?

L149 micron?

Results.

L191 "Tough note. . . " could be in parenthesis.

L196 "given distributions"?

L199 "likely"?

L214 "enhanced"?

L218 manuscript or article?

L234 Please add the total diversity to the figure on sensitivity. The pattern looks similar to the thermal Norm one and thus it seems to suggest that processes that impact the Thermal Norm diversity (notably, transport here) can be very important in setting the total diversity.

L387 Possibly? Several time in the text there are statements that are too strong. This is the case also for the comment on Lima-Mendez et al. The authors of this MS maybe right but they have no direct evidences to oppose. They can only suggest or hypothe-

size.

L392 and following: The limitation due to the low model resolution is never mentioned.

More importantly, as only briefly discussed at the very end of the Discussion, the trait-based modeling approach, while being much improved here, is still far from reproducing the observed richness (especially if quantified using genetic or genomic approaches). There are issues with data, indeed. But it is unclear from this manuscript which should the future directions of research based upon this kind of modeling approach.

Supplementary: please provide the main parameters values. Is the term in parenthesis in eq. S1.4 (1/T-1/TN) or actually (1/TN-1/T)?

––––––––––––––––––––––––

---

## Referee Comment (RC2) · A. C. Martiny (Referee) · 8 Nov 2019

Here, Dutkiewicz and colleagues using a biological rich model embedded in a global circulation model to examine underlying controls on global pattern of plankton alpha-diversity. Overall, it is a really nice study. It is well written, the results a clearly presented, the results are very interesting and the paper generally include a very thoughtful discussion. As such, I only have minor comments.

I really appreciate that the authors are very explicit about these results being found in a 'model' world. This distinction is often blurred.

Figure 1 is very convincing.

The study would benefit from a formal comparison between observations and model

outputs. Right now, we are left with a visual test. Most global ocean model studies suffer from this issue but I just don't like statements like 'similar pattern' and such. These statements sometimes cover an awful match. I don't this is the case here but nevertheless. . .

There is obviously a lot to learn from using an R* type framework. However, the framework (in general and as applied here) ignores a key ecosystem feature, whereby organisms switch between different variants of the same resource (e.g., ammonium, nitrite, nitrate, urea, other DON, etc.) – each likely less palatable. This possibility for resource substitution changes the dynamics of diversity in relation to nutrient levels. For instance, it is likely much harder to have competitive exclusion and specialization in one resource might come at the expense of others. This does not invalidate the current study in anyway and it would be challenging to model all these additional tracers. However, I think it would useful to discuss this limitation – especially as it relates to the emergent diversity patterns.

Do you have any issues with the smallest or largest size class? In other words, are there biological boundary problems due to less competition at the edges.

L449: I think it is a mistake to think of latitude as an environmental factor. Also, I think it is unfair to characterize past studies as simple statistical correlations. When people are looking for relationships to latitude, they are not arguing that plankton respond to where they are located on a map. Rather latitude is a placeholder for a range of abiotic and biotic interactions. Thus, I think it is reasonable to look at relationship with latitude and I found this section a tad too negative about past efforts.

---

## Author Comment (AC1) · 28 Nov 2019

The MS by Dutkiewicz et al. presents a detailed account on the drivers of the marine phytoplankton diversity in a numerical model.

The effort is of great interest since the model considers all the processes that are considered relevant when using a trait-based framework. It is also of interest since the impact the main traits and processes are discussed separately, using a set of well defined sensitivity experiments. Finally, it focusses the discussion on the immediate implications for the interpretation of real data. In particular it points out that the selection of the environmental variables that are used as explanatory variables in statistical analyses has to be coherent with hypotheses drawn from the current theories. Dramatically, if one uses a trait-based framework the outcome is that most of the variables are hard to be constrained quantitatively (eg, the nutrient fluxes).

Indeed, the actual phytoplankton richness is possibly orders of magnitudes higher than the one emerging from this model exercise. This limitation is possibly due to the strong limitations of the "classical" trait-based approach. In addition, in this exercise there are no significant conceptual novelties on specific processes. Nevertheless, as stated by the authors, a general synthesis of the lessons learnt using this framework is going to be very useful for future studies and, with given its pedagogic clarity, for students and young researchers.

I thus recommend it for publication after addressing some very minor points.

We thank the reviewer for the positive comments. We address these points below in blue text. We note that the diversity is indeed significantly lower than in the real world. This now stated this several times in the revised version. For instance, in the model description (lines 165, revised version 175):

*"We also emphasis that the level of richness that the model captures, though large for a model, is orders of magnitude lower than the real ocean. Thus this is not a fully comprehensive study of diversity, but does never-the-less provide a promising avenue for understanding some of the controls on diversity."*

In the model limitation section 6, lines 417 (new version lines 449-450):

*"Our model only captures a tiny (probably orders of magnitude less) amount of the diversity found in the real ocean. Including more resolution along these axes and including additional trait axes would allow for further diversity, but is beyond the scope of this present study."*

A check is required for all the citations (missing parenthesis or points).

In the revision we have checked for missing parentheses and citations. (And yes, we found several).

Specific comments:

Introduction. The last sentence is generally correct for the whole diversity but in most cases studies focus on single groups. Is it still true?

Yes, we believe this is still true even for a distinct group. For instance, if the group is diatoms, then our study suggests that transport will still be important for hotspots of diatom diversity, while size/species

specific losses and resource supply will dictate size diversity within diatoms. We did obliquely refer to this in lines 427-429:

*"Our results suggest that observed patterns of "total" diversity (or for any grouping of phytoplankton types, such as for nano and micro-eukaryotes along the AMT) are a result of multiple controllers: supply rate of limiting resource, imbalance in supply of different resources relative to competitor's demands, top-down control, particularly in terms of size-dependent grazing, and transport processes."*

Introduction: A statement on the different definitions of "diversity" is missing. A general issue with the literature on plankton is the lack of discussion about the importance and the technical and ecological implications of the choice of the metrics for diversity.

We do discuss what we mean by "diversity" in the context of the paper (lines 159-162). We also have a discussion about some of the techniques of measuring diversity in the Discussion (see lines 475-485). However, we agree with the reviewer that this also warrants a statement in the discussion (and also agree that there is a lack ok of such discussion in many studies on "diversity"). As such we have added the following at line 68:

*"In this study we will almost exclusively consider diversity in terms of "richness", the number of locally co-existing species. This definition is often referred to as alpha-diversity. We focus on richness here as the ecological theories we use explain co-existence, rather than other common metrics of diversity such as Shannon Index or evenness. Given the model setup, we also do not consider the rare biosphere."*

The study by Lima-Mendez is not on diversity but on interactions. Their conclusions that biotic interactions are more important than environmental factors in setting the community network derive from the analysis of a dataset that contains much more "species" than this model. I think it is just not possible to compare the two approaches with the current state of understanding. In addition, the model has no interactions except for grazing. Thus citing it is useful especially to discuss how these results represent a challenge for the current modelling approaches.

We agree that the Lima-Mendez paper is not about diversity, and as such we remove it from the Introduction. However, this is a valuable paper and we therefore do still cite it in the Discussion, but now make clearer that that paper was about community structuring rather than diversity. We did not mean to sound as though we were comparing the two approaches, and have changed the wording in the Discussion so that is no longer misleading (line 455)

*"In a study focusing on the interactions (and hence community structure) showed little statistical links to nutrient concentrations (e.g. Lima-Mendez et al., 2015)."*

L45 "there is evidence suggest"?

Now changed to "suggesting"

Section 2.

What is the definition of richness used for the AMT data? What is the reason for not using a rarefaction of the data prior to define richness? The issue should be discussed shortly, also considering the method used here (L94-95).

*The definition of richness in the AMT is now included in this section (see text quoted below). Since the model has to impose a threshold of abundance (or biomass) for defining presence/absence of population types, and thus for defining richness for the AMT we consider that the cleanest comparison is with raw species richness data rather than by using rarefaction (i.e. such that we do not encompass the rare species). We also make this assumption clearer in the text.*

*Near line 87 (revised text line 89-91):*

*"Here diversity is determined as richness, which in this study is defined as the number of species detected in sample volumes in the range 10-100 ml."*

*And after line 100 (revised version 105-106):*

*"Given how these data are compared to model output (see below) we purposely neglect the rare biosphere, so do not attempt any techniques such as rarefraction to account for the rare species."*

*And altered text around 159-165 (revised version 166-174):*

*"As mentioned in the introduction, in this study we primarily discuss diversity in term of "richness" defined here as the number phytoplankton types that co-exist at any location above a threshold. We, in particular, look at the annual mean of the instantaneous surface richness (though see Supplemental for examples with depth). Technically we use a threshold value ($10^{-5}$ mmolC/m$^3$) to determine if a type is in existence at any spot. This value would convert to about 10 Prochlorococcus cells/ml (typical oligotrophic waters are above $10^3$ cells/ml), or only a tiny fraction ($10^{-4}$) of a larger diatom cell/ml.  Thus this definition neglects the rare species that would be difficult to separate from numerical noise. This is why we do not account for the rare species in the AMT observations discussed above."*

The model resolution is very low for the current standards for the ocean physics. Presumably, the computational requirements to run the biogeochemical model are such that using a higher resolution was too demanding. Nevertheless, in discussing the limits of the study the lack of mesoscale and submesoscale processes should be mentioned.

*We do mention the coarse resolution of the model (lines 185-187 and lines 387-388). But agree that this is not sufficiently discussed as a limitation. We now include additional text and feel that this significantly improves this article.*

*Near line 113 (revised version 119-121):*

*"At this horizontal resolution, the model does not capture mesoscale features such as eddies and sharp fronts, a limitation of the model that must be kept in mind when considering the results."*

And near line 354 (revised version lines 383-385):

*"Both Clayton et al (2013) and Levy et al (2014) showed the importance of eddies in enhancing this process of transport mediated diversity. Thus the hotspots in the default experiment would likely be even higher in a model that did resolve the mesoscale."*

And also section 6 (Limitation of this study), after line 420 (revised version lines 455-460):

*"Given computational constraints with this complexity of ecosystem model, we have use a coarse resolution physical model that does not capture explicit meso (or sub) scale features. Previous studies (e.g. Clayton et al 2013; Levy et al 2014) have shown the importance of such features in enhancing diversity. Mesoscale features are important in temporal increases in nutrient supplies (see e.g. Clayton et al., 2017), and from this study this suggests temporal increase in size classes during such events. Sub- and mesoscale mixing in frontal regions will also enhance the richness in hotspots (Clayton et al 2013), but also in a general increase richness (Levy et al 2014)."*

L146 Missing the verb?

Not a verb, but rather a qualifies. Thanks for catching this. Now reads:

*"Following empirical evidence, mixotrophic dinoflagellates are assumed to have lower maximum photosynthetic growth rates than other phytoplankton of the same size (Tang, 1995; Fig 4a) and lower maximum grazing rates than heterotrophic dinoflagellates of the same size (Jeong et al., 2010, Supplemental Fig S2)."*

L149 micron?

Yes, now changed

Results. L191 "Though note. . . " could be in parenthesis.

Agreed, this has been done in the revised version

L196 "given distributions"?

Thanks, this was a typo, Text is now altered to say: "compared to"

L199 "likely"?

Yes – large classes are definitely under-estimated. In revised text "likely" is removed.

L214 "enhanced"?

Changed to "enhanced"

L218 manuscript or article?

We've change to "study" to be consistent to the rest of the article.

L234 Please add the total diversity to the figure on sensitivity. The pattern looks similar to the thermal Norm one and thus it seems to suggest that processes that impact the Thermal Norm diversity (notably, transport here) can be very important in setting the total diversity.

We are a little confused here, as Figure 8 (which is discussed in the paragraph starting at line 234) does have the total diversity. Though at a quick glance the total and thermal norm richness looks similar, the total is indeed made up of all the dimensions.  To avoid this confusion, we now add at lines 241 (revised text 260-263):

*"At first glance total diversity (Fig 8a) may look most like the thermal norm diversity (Fig 8d), but this is mostly because our eyes are drawn to the hotspots. In reality total diversity patterns are strongly impacted by all three dimensions of diversity as will be shown more clearly by the sensitivity experiments discussed later."*

Perhaps the reviewer is suggesting adding the total diversity to Fig 11 (the sensitivity experiments)? We agree that this is a good idea. The revised version of the figure has the total (see below). This is a rather nice illustration of how the diversity decreases in all sensitivity experiment and that thermal norm diversity is not the same pattern as the total, so we have added additional text.

(revised version 317-318):

*"However, the total diversity reduces dramatically (Fig 11, top row). Patterns of hot spots are however still apparent, but the increases in diversity with higher nutrient supply is no longer apparent."*

after line 303 (revised version lines 366):

*"Total diversity is reduced everywhere, but mostly in the lower latitudes where the loss of diazotrophs and coccolithophores has a high impact."*

Line 354 (revised version line 382-385):

*"Total diversity is reduced everywhere, but most dramatically in these hotspot regions."*

L387 Possibly? Several time in the text there are statements that are too strong. This is the case also for the comment on Lima-Mendez et al. The authors of this MS maybe right but they have no direct evidences to oppose. They can only suggest or hypothesize.

We have revised the text to emphasize where we can only hypothesize. We have removed the mention of Lima-Mendez in the introduction and have clarified our statement of this article in the discussion. For instance Line 446-456 (revised version 488-500):

*"Though observational studies have hypothesized a multi-factorial control on diversity in the ocean (e.g. Rodriquez-Ramos et al 2015), they were unable to find significant correlations with any combination of factors such as latitude, temperature or biomass, or even nutrient concentrations. Correlating with factors such as temperature, latitude is a logical first step for trying to understand observed patterns of diversity, as these are often the only additional data that is available from a field study, and for instance "latitude" could potentially stand in for a range of biotic and abiotic processes. Our study, however, suggests that to some degree these factors are unlikely to help disentangle controllers of diversity. For instance, in our study it is mixing of different temperature water masses, potentially hinted at by local temperature variances rather than temperature itself, that is important. In a study focusing on the interactions (and hence community structure) showed little statistical links to nutrient concentrations (e.g. Lima-Mendez et al., 2015). On the other hand nutrient supply rates (a harder variable to measure) did show some measure of identifying communities (see e.g. Mouriño-Carballido et al. 2016)."*

In other parts of the text we have added qualifiers or removed sentences that we, on hindsight, deem to be too strongly stated.

L392 and following: The limitation due to the low model resolution is never mentioned.

Yes, this was an oversight. As discussed above, we have now added several sentences on this issue in several parts of the articles, in particular in this section, after line 420 (revised version lines 455-460):

*"Given computational constraints with this complexity of ecosystem model, we have use a coarse resolution physical model that does not capture explicit meso (or sub) scale features. Previous studies (e.g. Clayton et al 2013; Levy et al 2014) have shown the importance of such features in enhancing diversity. Mesoscale features are important in temporal increases in nutrient supplies (see e.g. Clayton et al., 2017), and from this study this suggests temporal increase in size classes during such events. Sub- and mesoscale mixing in frontal regions will also enhance the richness in hotspots (Clayton et al 2013), but also in a general increase richness (Levy et al 2014)."*

More importantly, as only briefly discussed at the very end of the Discussion, the traitbased modeling approach, while being much improved here, is still far from reproducing the observed richness (especially if quantified using genetic or genomic approaches). There are issues with data, indeed. But it is unclear from this manuscript which should the future directions of research based upon this kind of modeling approach.

We have added the following sentences to address this issue, starting in the introduction, (lines 165, revised version 175):

*"We also emphasis that the level of richness that the model captures, though large for a model, is orders of magnitude lower than the real ocean. Thus, this is not a fully comprehensive study of diversity or species richness, but does never-the-less provide a promising avenue for understanding some of the controls on diversity."*

In the model limitation section 6, lines 417 (new version lines 449-450):

*"Our model only captures a tiny (probably orders of magnitude less) amount of the diversity than is in the real ocean. Including more resolution along these axes and including additional trait axes would allow for further diversity, but is beyond the scope of this present study."*

Supplementary: please provide the main parameters values. Is the term in parenthesis in eq. S1.4 (1/T-1/TN) or actually (1/TN-1/T)?

The phytoplankton allometric parameters are already given in Table 1 of the supplement (now Supplemental Table 2). We now add a new Supplemental Table 1(see below) which includes the values for all the other parameters mention in the supplemental text. We now direct the reader to these tables in the revised Supplemental Material). We feel that it would be confusing to include all the other model parameters not mentioned in the text since we could not adequately explain these. We however direct the reader to Dutkiewicz et al (2015) which has all the equations and all the parameters values listed. Almost all parameter values used here are identical to those used in that study. We have included a new section in the Supplemental (revised version Section S1.4: Model Parameters) where we explain more clearly where to find the appropriate parameters (e.g. Dutkiewicz et al (2015) Table 1 and 2 and those in our previous study), and detail the very few parameters that have been changed from Dutkiewicz et al 2015.

Equation S1.4 is correct.

New section in supplement:

*"**S1.4. Model Parameters:** We provide the values for the non-allometric parameters mentioned in the text above in Supplemental Table 1 and for the allometric parameters in Table 2. We refer the reader to Dutkiewicz et al (2015a) Tables 1 and 2 for the values of all other ecological and biogeochemical parameters used in this model. We note here only the few changes in parameter values: In Dutkiewicz et al (2015a) we had preferential remineralization of dissolved organic phosphorus (DOP) relative to other elements, here we do not. In this study, DOP remineralizes with same values (0.0333 $d^{-1}$) as the other elements. We found that CDOM was too high in this version of the model and increased the CDOM bleaching rate to 0.2592 $d^{-1}$ from 0.167 $d^{-1}$."*

[Figure]

**(Revised) Figure 11: Sensitivity simulations, model annual mean richness**. EXP-1 has no size-dependent loss rates (i.e. only one grazer); EXP-2 has no nutrient requirement differences between functional groups; EXP-3 has no transport of the plankton (all nutrients and non-living organic pools are still transported). Top row: total richness; Second tow: size class richness determined by number of co-existing size classes; Thirdrow:  functional richness determined by number of co-existing biogeochemical functional groups; Bottom row: thermal richness determined by number of co-existing temperature norms. The left most column are the same output as shown in Fig 9a,b,c,d for the original ("default") experiment, but with absolute values, not normalized.

|  | Symbol | Value | Units |
|---|---|---|---|
| normalization factor for temperature function | $\tau_T$ | 0.8 | unitless |
|  | $A_T$ | -4000 | K |
| reference temperature | $T_N$ | 293.15 | K |
| factor determining width of norms | $B_T$ | $3 \times 10^{-4}$ | 1/K |
| norm optimum temperature | $T_{oj}$ | 271.15 to 304.15 in 4K intervals | K |
| decay coefficient for norms | $B$ | 4 | unitless |
| palatibility matrix | $\sigma_{jk}$ | 1 if grazer $k$ is 10 times larger the prey $j$. | unitless |

| | | 0.3  if grazer $k$ is 5 or 15 times larger than prey $j$ | |
|---|---|---|---|
| grazing half saturation rate | $k_p$ | 1.5 | mmolC/m³ |

**Supplemental Table S1: Non-allometric ecological parameters mentioned in this Supplemental**

---

## Author Comment (AC2) · 28 Nov 2019

Here, Dutkiewicz and colleagues using a biological rich model embedded in a global circulation model to examine underlying controls on global pattern of plankton alphadiversity. Overall, it is a really nice study. It is well written, the results a clearly presented, the results are very interesting and the paper generally include a very thoughtful discussion. As such, I only have minor comments.

We thank the reviewer for these positive comments and appreciate the improvements that they make to the article. Below which we respond to the reviewer's comments (black text) in blue text.

I really appreciate that the authors are very explicit about these results being found in a 'model' world. This distinction is often blurred.

Thank you. It is good to maintain this distinction, but also to show how insight from the model can be applied to the real world.

Figure 1 is very convincing.

The study would benefit from a formal comparison between observations and model outputs. Right now, we are left with a visual test. Most global ocean model studies suffer from this issue but I just don't like statements like 'similar pattern' and such. These statements sometimes cover an awful match. I don't this is the case here but nevertheless. . .

We now include a new section in the supplemental with more formal evaluation (biases, spatial correlation and standard deviations) of the model against a variety of satellite and in situ observations (Revised version Supplemental Section S2 and new Supplemental Figs S3-S8). This includes both global level and also more explicit evaluation against the AMT data. However, we note that it is difficult to make a clean comparison between snapshots from the cruises and the model. We describe this more in the Revised Supplemental Section S2. We believe that the figures (e.g. Fig 1 and 5) in the main text are much clearer as they are (rather than the figures designed to show the bias explicitly, Supplemental Figs S6,S7), and plan to leave these as is in the main text. We do point the reader to this more formal evaluation in the revised version of the paper (Near lines 185, revise version lines 196):

*"Model development was guided by evaluating against a range of in situ and satellite-derived observations (see Supplemental text S2 and Supplemental Figures S3-S8). We refer the reader to the fuller evaluation in the Supplemental, but provide a brief version here."*

At the end of this response, please find the proposed new Supplemental section and figures.

There is obviously a lot to learn from using an R* type framework. However, the framework (in general and as applied here) ignores a key ecosystem feature, whereby organisms switch between different variants of the same resource (e.g., ammonium, nitrite, nitrate, urea, other DON, etc.) – each likely less palatable. This possibility for resource substitution changes the dynamics of diversity in relation to nutrient levels. For instance, it is likely much harder to have competitive exclusion and specialization in

one resource might come at the expense of others. This does not invalidate the current study in anyway and it would be challenging to model all these additional tracers. However, I think it would useful to discuss this limitation – especially as it relates to the emergent diversity patterns.

This is an interesting comment. We believe that the R* framework could be altered to address this issue, and it would be interesting to see what that would suggest for diversity, especially if different species had more/less affinity for any of the variants of some resource. This is beyond the scope of this current study.

However, importantly, the numerical model does include ammonium, nitrite, nitrate, with phytoplankton preferentially consuming ammonium. The insight from the R* framework as provided here does still help us understand the results even from this more complex system (ie. the model). We make this clearer in the text (near line 274, revised version lines 294-297)

*"We note that the model is significantly more complex than the simple theoretical framework, including multiple limiting nutrients, multiple variants of one of those resources ($NH_4$, $NO_2$ and $NO_3$) with differing affinities, additional loss terms (e.g. sinking) as well as more complicated grazing and foodweb (rather than food chain). However, this framework still helps us understand the patterns of size diversity in the model."*

Do you have any issues with the smallest or largest size class? In other words, are there biological boundary problems due to less competition at the edges.

The model does not capture as many larger size classes as observed. This is likely because there are other traits (shape, chain formation, buoyancy control) that we do not include. This is noted in the text (lines 200-202, revised version 215):

*"The model captures biomass in almost all size classes (Fig 6, Supplemental Fig S10a), though the largest size classes are likely underestimated. Traits not included in the model (e.g. buoyancy regulation, chain formation, symbiosis) are possibly more important for maintaining these large size classes."*

But now also mention in Section 6 (Limitation of this Study), near Lines 407 (revised version lines 436)

*"The model considers only three axes of phytoplankton traits. We anticipate that additional axes such as morphology (e.g. shape, spines), motility (e.g. flagella), chains, colony formation, nutrient storage abilities, and symbiosis will each have their own controlling mechanisms. Such traits might allow the model to capture more species, and particularly, more larger types."*

L449: I think it is a mistake to think of latitude as an environmental factor. Also, I think it is unfair to characterize past studies as simple statistical correlations. When people are looking for relationships to latitude, they are not arguing that plankton respond to where they are located on a map. Rather latitude is a placeholder for a range of abiotic and biotic interactions. Thus, I think it is reasonable to look at relationship with latitude and I found this section a tad too negative about past efforts.

We agree that it is potentially confusing to call latitude an "environmental" factor and have altered the text in many locations to reflect this. And we agree that latitude is often used to represent many other

factors. However, we argue that there are many different nutrient supply rates, and different levels of mixing along longitude at any given latitude. And as such we suggest latitude will not be able to fully explain diversity patterns. We have revised the text to elaborate on this point and to be a little less negative. We do not quote all revised text on these issues, but include an example here (line 448, revise version 490):

*"Correlating with factors such as temperature, latitude is a logical first step for trying to understand observed patterns of diversity, as these are often the only additional data that is available from a field study, and for instance "latitude" could potentially stand in for a range of biotic and abiotic processes. Our study however suggests that to some degree these may not be able to help disentangle controllers of diversity."*

**NEW SUPPLEMENTAL MATERIAL:**

**S2. Model Evaluation**

We evaluate the model against a range of in situ and satellite-derived observations (Main text Figs 1,5,7, and Supplemental Figs S3-S8). The model captures the patterns of low and high surface nutrients seen in the compilation of in situ observation from World Ocean Atlas (Garcia et al., 2014, Supplemental Fig S3). Nitrate is slightly too high in the Pacific gyres and too low along the equator. This reflects that iron limitation may be too strong in this region. But the correlation to observations is good (Supplemental Fig S5). Phosphate has similar, but accentuated, biases in the Pacific Equatorial region, and is also too high in the Southern Ocean. Phosphate is thus more evenly distributed than observed (Supplemental Fig S5). Likely the fixed stoichiometry of the model leads to phosphate concentrations not being sufficiently biologically modulated. Silicic acid also shows similar biases in the Equatorial Pacific and is too high in the Southern Ocean. This latter bias is likely a reflection of constant Si:C we impose. In the Southern Ocean, diatoms are more highly silicified (Tréguer et al 2017). This overestimation in the Southern Ocean leads to a higher spatial standard deviation relative to the observations (Supplemental Fig S5).

Chl-a compares well to satellite estimate (Supplemental Fig S4, S5). Note that the satellite estimates have large uncertainties (Moore et al, 2009 estimates more the 35% errors) and, moreover, the values shown for the satellite Chl-a estimates in Supplemental Fig S4 are not true annual means, but rather compilations of all available data, missing values when there are clouds or the light levels are too low (e.g. polar winters). The coarse resolution of the model does not capture important physical processes near coastlines, and lack of sedimentary and terrestrial supplies of nutrients and organic matter lead to Chl-a being too low in these regions. Chl-a is under-estimated by the model in the subtropical gyres, likely due to lack of mesoscale processes in the model that would supply additional nutrients in these regions (see e.g. Clayton et al 2017). The model Chl-a is higher than the satellite estimates in the high latitudes. Regional biases in the satellite algorithms are likely, particularly an issue in the Southern Ocean (e.g. Szeto et al., 2011, Johnson et al. 2013). The model though has a good correlation with the observations and captures the spatial variability well (Supplemental Fig S5).

We further compare the model to satellite-based estimates of Chl-a in different size classes (Main Text Fig 7, Supplemental Fig S4, S5), using the product from Ward et al (2005). Here we capture the ubiquitous pico-phytoplankton and the limitation of the larger size classes to the more productive regions. The model pico-phytoplankton size class Chl-a is potentially slightly too low and the nano size class too high. Though we note that if we set the pico/nano break at the model 5$^{th}$ size class (just under 3μm) instead at the 4$^{th}$ (2μm) size class, the relative values are much more in line with the satellite product. We suggest that the satellite product division might not be that exact. The micro-size class matches in location to the satellite product but is slightly too low as discussed above, but has the least impressive correlation to the observations (Supplemental Fig S5).

We also compare the model functional group distribution to the latest compilation of observations (Main Text Fig 7b, MAREDAT, Buitenhuis et al 2013, and references therein). The observations are sparse and here we average all observations regardless of season in 5 degree bins (Main Text Fig 7b). With such spatially and temporally sparse observations, we do not believe it makes sense to calculate biases or correlations between the model and observations, and we rely on visual evaluation. Though the observations are sparse, we do capture the ubiquitous nature of the pico-phytoplankton, the limited domain of the diazotrophs (including observed lack of diazotrophs in the South Pacific gyre), the pattern of enhance diatom biomass in high latitude, and low in subtropical gyres. We over-estimate the coccolithophore biomass relative to MAREDAT in many regions, but note that the conversion from cells to biomass in that compilation was estimated to have uncertainties as much as several 100% (O'Brien et al., 2013). The MAREDAT compilation did not include a category for dinoflagellates.

We further evaluate the model against the in situ observations as captured during the Atlantic Meridional Transects (AMT) 1,2,3, and 4 (Main Text Fig 1, 5, Supplemental Figs S6,S7,S8). AMT2 and 4 occurred during April and May of consecutive years, while 1 and 3 took place during September and October. Here we compare the range of values found in the two cruises in each time period to the range of values in the model during the two-month period (Supplemental Figs S6,S7). Similar to the global evaluation above, we find that silicic acid is too high in the Southern Ocean (Supplemental Fig S6) and that Chl-a is underestimated in the subtropical gyres. We note that the model Chl-a compares better to the Southern Ocean in situ observations than they do to the satellite estimates. Though the correlation is reasonable, the spatial variability is too low (Supplemental Fig S8a,b). The phytoplankton functional groups compare less well to observations than the nutrients and Chl-a, but are still plausible. Coccolithophore biomass however drops too low in the Southern Ocean, likely due to the model smallest diatom being parameterized as too competitively advantaged. However, pleasingly, the relative abundances of the three groups (diatoms, coccolithophores and dinoflagellates) are captured: Diatom biomass is much lower in the subtropical gyres than the other two functional groups, and higher in the Southern Ocean and coccolithophores and dinoflagellates as having much more even distributions.

As a final model evaluation, we compare the model estimates of richness against those found along the AMT (Main Text Fig 1, Supplemental Fig S7, S8b,c). As expected, given the only 350 species parameterized in the model, the model has lower diversity than seen in the AMT. But, the model does captures the low and high patterns of total richness along the AMT (Supplemental Fig S7a,d), though underestimates the diversity in the subtropical gyres. In these regions it is likely that traits axes (e.g. symbiosis, colony formation etc) not captured in the model provide additional means for phytoplankton to co-exist. The richness within different functional groups is also captured, though much better for diatoms than the other two groups (Supplemental Figs 7b,e, Supplemental S8c,d). Excitingly the model also captures the

differences in the diversity within functional groups and in size classes. Diatoms have much larger diversity in the Southern Ocean than the other functional groups, while coccolithophores and mixotrophic dinoflagellates diversity is much more uniform across the transect. AMT richness was also calculated for different size classes. The model does well in capturing these divisions as well (Supplemental Fig S7c,f, S8c,d). The model captures the much higher diversity within the smallest size category (2-10$\mu$m) and the lower and much more regionally varying diversity in the larger size category, including the lack of diversity in the largest size class (>20$\mu$m) in the subtropical gyres.

**NEW SUPPLEMENTAL REFERENCES**

Clayton, S. Dutkiewicz, O. Jahn, C.N. Hill, P. Heimbach, and M.J. Follows.  Biogeochemical versus ecological consequences of modeled ocean physics. *Biogeosciences*, **14**, 2877-2889 (2017).

Garcia, H. E., R. A. Locarnini, T. P. Boyer, J. I. Antonov, O.K. Baranova, M.M. Zweng, J.R. Reagan, D.R. Johnson, 2014. *World Ocean Atlas 2013, Volume 4: Dissolved Inorganic Nutrients (phosphate, nitrate, silicate).* S. Levitus, Ed., A. Mishonov Technical Ed.; NOAA Atlas NESDIS 76, 25 pp.

Johnson, R., Strutton, P.G., Wright, S.W., McMinn, A., and Meiners, K.M.: Three improved Satellite Chlorophyll algorithms for the Southern Ocean, J. Geophys. Res. Oceans, 118, doi:10.1002/jgrc.20270, 2013

Moore, T.S., Campell, J.W., and Dowel, M.D.: A class-based approach to characterizing and mapping the uncertainty of the MODIS ocean chlorophyll product. Remote Sensing of Environment, 113, 2424–2430, 2009.

Szeto, M., Werdell, P.J., Moore, T.S., and Campbell, J.W. :Are the world's oceans optically different?, J. Geophys. Res., 116, C00H04, doi:10.1029/2011JC007230, 2011.

Tréguer, P., Bowler, C., B. Moriceau, S. Dutkiewicz, M. Gehlen, K. Leblanc, O. Aumont, L. Bittner, R. Dugdale, Z. Finkel, D. Iudicone, O. Jahn, L. Guidi, M. Lasbleiz, M. Levy, and P. Pondaven. Influence of diatoms on the ocean biological pump. *Nature Geoscience*, doi:10.1038/s41561-017-0028-x (2017).

**NEW SUPPLEMENTAL FIGURES**

[Figure]

**Supplemental Figure S3: Annual Mean Surface (0-10m) Nutrients**. (Top row) Nitrate (mmolN/m³); (Middle row) Phosphate (mmolP/m³ ); (Bottom row) Silicic acid (mmolSi/m³). (Left column) Model, 5th year annual mean; (Middle column) Observations , annual climatology, from World Ocean Atlas (Garcia et al 2013); (Left Column) Model bias determined as model minus observation.

[Figure]

**Supplemental Figure S4: Annual Chl (mgChl/m³)**. (Top row) total Chl-a; (Second row) Chl in micro (>20µm) size class; (Third row) Chl in nano (2-20µm) size class; (Bottom row) Chl in pico (<2µm) size class. (Left column) Model, 5th year; (Middle Column) Satellite Observations, top from NASA MODIS; other three panels are the satellite based estimates from Ward (2005); (Right Column) Model bias determined as model minus observations. The middle column shows annual "climatology" of all available satellite measurements, with missing observations in the polar winters; while model results are annual mean (0-10m).

[Figure]

**Supplemental Figure S5: Taylor Diagram of Global Annual Surface Fields**. This polar coordinate plot shows correlation (angular position) and the normalized (by observed spatial STD) spatial standard deviation (radial position) between model and observation for the fields shown in Supplemental Figures S3 and S4. Statistics are performed on log-normalized fields. REF indicates a perfect match between model and observations. NO3, PO4, SIL refer to nitrate, phosphate and silicic acid respectively; Observations are from World Ocean Atlas (Garcia et al 2014). CHL refers to total Chl-a; Observations are satellite estimates from NASA MODIS. Mic, Nan, Pic refer to Chl-a in the micro (>20μm), nano (2-20μm), pico (<2μm) respectively; Observations are the satellite-based estimates in each size class from Ward et al (2005).

[Figure]

**Supplemental Figure S6: Atlantic Meridional Transect Model and In situ Observations**. Left Column is for April/May (AMT2,4) results, Right Column for September/October (AMT1,3). Circles indicates average of the two AMT cruises in 4° latitude bins in each time period, and the vertical line across each circle shows the range of the observations. Solid lines indicate the model two-month mean and dashed lines indicate the model minimum and maximum from that two-month period. (a), (b) surface nutrients (black=nitrate, mmolN/m³; green=phosphate, 16xmmolP/m³; light blue=silicic acid, mmolSi/m³); (c), (d) surface Chl-a (mg Chl/m³); (e), (f) surface phytoplankton biomass (mg C/m³); red=diatoms; blue=coccolithophores; purple=dinoflagellates).

[Figure]

**Supplemental Figure S7: Atlantic Meridional Transect Model and In situ Observations of richness**. Left Column is for April/May (AMT2,4) results, Right Column for September/October (AMT1,3). Circle indicates average of the two AMT cruises in each time period in 4° latitude bins, and the vertical line across each circle shows the range of the observations. Solid lines indicate the model two-month mean and dashed lines indicate the model minimum and maximum from that two-month period. Normalized richness of (a),(d) all diatoms, coccolithophores and dinoflagellates together; (b),(e) each functional groups separately (red: diatoms, dark blue: coccolithophores, purple: dinoflagellates); (c),(f) 3 size classes (light blue: 2-10μm, black: 10-20μm, green: >20μm). Model pico-phytoplankton and diazotrophs are not included in the model analysis as they were not analyzed in the observations.

[Figure]

**Supplemental Figure S8: Taylor Diagram of Atlantic Meridional Fields**. This polar coordinate plot shows correlation (angular position) and the normalized (by observed spatial STD) spatial standard deviation (radial position) between model and observation for the fields shown in Supplemental Figures S6 and S7. Left Column is for April/May (AMT2,4) results, Right Column for September/October (AMT1,3). We compare the in situ two-cruise mean (circles in Fig S6 and S7) against the model two-month average (solid lines) averaged onto the same 4° latitude bins. REF indicates a perfect match between model and observations. (a),(b) NO3, PO4, SIL refer to nitrate, phosphate and silicic acid respectively. CHL refers to Chl-a. DIA, COC, DIO refer to diatom, coccolithophore and dinoflagellate biomass respectively. Statistics are performed on log-normalized fields for the Chl-a and biomass fields. (c),(d) normalized richness where TOT refers to the total richness DIA, COC, DINO refers to the richness in diatoms, coccolithophores, and dinoflagellates respectively, and SMA, MED, LAR to the 3 size classes (2-10µm, 10-20µm, >20µm) respectively.

---

## Author Response (AR2)

We thank the editor for his careful reading of our paper. All the corrections listed below have been included in the latest uploaded pdf.

Stephanie Dutkiewicz

At multiple places (Lines 39, 51, 258, 410) you use 'on the other hand' without 'on the one hand'. Please rephrase to However, In contrast or alike.

All the above "on the other hands" have been removed and replaced with more appropriate wording.

Line 228: diazotrophs

Corrected

Line 286: required for it to …

Corrected

Line 366: k-strategists

Corrected

In the appendix space after symbols are sometimes missing: line 583, 585, 587.

Corrected

Line 584: insight: with higher…

Corrected

Line 965: eutrophied

Corrected